# Complexity is not Enough for Randomness

Shiyong Guo, Martin Sasieta, Brian Swingle

*Martin Fisher School of Physics, Brandeis University*

*Waltham, Massachusetts 02453, USA*

## Abstract

We study the dynamical generation of randomness in Brownian systems as a function of the degree of locality of the Hamiltonian. We first express the trace distance to a unitary design for these systems in terms of an effective equilibrium thermal partition function, and provide a set of conditions that guarantee a linear time to design. We relate the trace distance to design to spectral properties of the time-evolution operator. We apply these considerations to the Brownian $p$-SYK model as a function of the degree of locality $p$. We show that the time to design is linear, with a slope proportional to $1/p$. We corroborate that when $p$ is of order the system size this reproduces the behavior of a completely non-local Brownian model of random matrices. For the random matrix model, we reinterpret these results from the point of view of classical Brownian motion in the unitary manifold. Therefore, we find that the generation of randomness typically persists for exponentially long times in the system size, even for systems governed by highly non-local time-dependent Hamiltonians. We conjecture this to be a general property: there is no efficient way to generate approximate Haar random unitaries dynamically, unless a large degree of fine-tuning is present in the ensemble of time-dependent Hamiltonians. We contrast the slow generation of randomness to the growth of quantum complexity of the time-evolution operator. Using known bounds on circuit complexity for unitary designs, we obtain a lower bound determining that complexity grows at least linearly in time for Brownian systems. We argue that these bounds on circuit complexity are far from tight and that complexity grows at a much faster rate, at least for non-local systems.

shiyongguo@brandeis.edu
martinsasieta@brandeis.edu
bswingle@brandeis.edu

# 1    Introduction

The Hilbert space provides a largely redundant description of the set of physically accessible states of a conventional many-body quantum system. Namely, starting from some physical reference state, it is impossible for a system to dynamically explore the vast majority of the Hilbert space efficiently restricted to few-body interactions, given that such class of Hamiltonians can only generate a polynomial quantum computation [1].

Still, it is interesting to consider more general systems governed by Hamiltonians with a larger degree of non-locality, systems for which the Hilbert space as a whole could make sense. In general, given some Hilbert space $\mathcal{H}$ of dimension $d$, any two pure states $|\psi_1\rangle, |\psi_2\rangle \in \mathcal{H}$ can be efficiently connected by a large class of non-local Hamiltonians. For instance, for mutually orthogonal states, explicit Hamiltonians which perform this task can be easily written down,

$$H = |\psi_2\rangle\langle\psi_1| + |\psi_1\rangle\langle\psi_2| + H_\perp \,, \tag{1.1}$$

where $H_\perp$ acts trivially on the subspace generated by $|\psi_1\rangle$ and $|\psi_2\rangle$. To set the time scale, we require the Hamiltonian to be normalized such that $\mathrm{Tr}(H^2) = d$.[1] The time evolution with (1.1) generates $|\psi_2\rangle$ from $|\psi_1\rangle$ in a time $t = \frac{\pi}{2}$. The form of (1.1) can be trivially generalized to the case of non-orthogonal states, which is the case with unit probability if $|\psi_2\rangle$ and $|\psi_1\rangle$ were to be relatively random. In this sense, all of $\mathcal{H}$ becomes physical if non-local Hamiltonians are allowed.

To set up the terminology, from now on we shall refer to conventional many-body systems with few-body interactions as *p-local systems*, where $p$ denotes the degree of locality of the Hamiltonian in some natural factorization of the Hilbert space, and is always taken to be fixed as the system is scaled.[2] On the other hand, we shall refer to systems which are not $p$-local as *non-local systems*, without particular reference to the precise scaling of the degree of locality of the Hamiltonian.

Non-local systems have been proven really useful as analytic models of structured chaotic Hamiltonians; an observation that originated with the study of heavy atomic nuclei [2–5]. At early times, however, the behavior of $p$-local and non-local systems is significantly different. Non-local systems generally scramble localized information – assuming such a notion even exists – parametrically faster than the fastest conventional chaotic systems, which are expected to correspond to black holes [6–11]. Moreover, non-local systems are believed to behave as 'hyperfast quantum computers', in the sense that the time-evolution in these systems generally corresponds to an exponentially powerful quantum computation, one which cannot be reproduced efficiently by any conventional quantum computer [12–16]. Some of these significant differences become irrelevant at the level of the late time physics where $p$-locality is less manifest. For example, in both cases nearby eigenvalues of time-independent Hamiltonians tend

---

[1] This normalization simply fixes the variance of the Hamiltonian in the maximally mixed state, and the eigenvalues of $H$ generally correspond to $O(d^0)$ numbers.

[2] Throughout this paper, locality refers to the number of degrees of freedom that interact in a given term in the Hamiltonian, not to any notion of geometric locality.

to repel each other and the unfolded energy spacings in the spectrum are statistically determined by a Wigner-Dyson distribution. This common feature is what defines quantum chaos in conventional systems at the level of the spectrum [2–5].

In this paper, we study the role of $p$-locality in the dynamical generation of unitary randomness in chaotic many-body quantum systems. We identify a surprising common feature between $p$-local and non-local systems, when it comes to the dynamical generation of unitaries distributed according to the Haar measure. Intuitively, $p$-local systems are not expected to be efficient unitary randomness generators, given that Haar random unitaries are exponentially complex to implement on a quantum computer. However, we find that the non-local systems that we study in this paper, albeit being quite generic, also fail to generate randomness efficiently. We can informally state our main conjecture, which posits the full generality of our results, in the form:

*Most non-local time-dependent Hamiltonians fail to efficiently generate Haar random unitaries.*

Let $\mathsf{U}(d)$ denote the space of unitary transformations acting on $\mathcal{H}$. More precisely, we conjecture that for most time-dependent Hamiltonians the time-evolution operator

$$U(t) = \mathsf{T}\,\exp\left(-i\int_0^t \mathrm{d}s H(s)\right),\tag{1.2}$$

is distinguishable – in a sense which will be made precise later – from typical unitary drawn from the Haar distribution on $\mathsf{U}(d)$ at polynomial times $t = \mathrm{poly}(\log d)$. As a consequence, the Haar distribution of unitaries in $\mathsf{U}(d)$ cannot be generated dynamically in any realistic timescale, and might be regarded as effectively unphysical in all cases, even if highly non-local systems were to exist.

As an example, we can consider the Hilbert space $\mathcal{H}$ of $N$ qubits such that $d = 2^N$. The failure of efficiency here means that even if we consider a fully non-local time-dependent random Hamiltonian with a natural normalization, it still takes a time which is superpolynomial in $N$ to generate unitaries which well-approximate all moments of the Haar distribution. This is the motivation for our focus on polynomial (efficient) versus exponential (inefficient) scaling with $N \propto \log d$, which is the entropy of the maximally mixed state on the Hilbert space.

An obvious objection to our conjecture comes from the fact that for any unitary, there are explicit choices of time-independent Hamiltonians which generate it efficiently. This applies to Haar random unitaries as well. Say we pick a unitary $U_{\mathrm{Haar}}$ sampled randomly from the Haar distribution. This unitary can indeed be reached in a unit of time with the time-independent Hamiltonian

$$H_{\mathrm{Haar}} \equiv i \log U_{\mathrm{Haar}} \quad \Rightarrow \quad U_{\mathrm{Haar}} = e^{-iH_{\mathrm{Haar}}}.\tag{1.3}$$

where the eigenvalues of this Hamiltonian are defined in the interval $(-\pi, \pi]$ so that the logarithm has the usual branch cut. The Hamiltonian (1.3) is correctly normalized up to $O(d^0)$ proportionality constants with extremely high probability, given that the eigenvalues of a Haar random unitary are

approximately uniformly distributed phases. Therefore, we have constructed, for any Haar random unitary, a time-independent Hamiltonian which generates it efficiently.

However, we claim that Hamiltonians like (1.3), or time-dependenent generalizations, which are able to generate Haar random unitaries efficiently, are either: $i$) highly atypical in any reasonable ensemble of Hamiltonians, and due to the concentration of the measure defining the ensemble at large values of $d$, are exponentially unlikely (in $d$) to be generated, or $ii$) arise typically in ensembles of Hamiltonians which are extremely concentrated and violate universal properties of random matrices.

A way to see this for $H_{\mathrm{Haar}}$ in (1.3) is to consider its spectrum. Recall that the nearby eigenvalues of a random Hamiltonian repel each other, and the spectrum behaves as a one-dimensional Coulomb gas confined in a potential. The potential determines the particular ensemble of Hamiltonians. In a mean field approximation, the spectral density can be determined from the static equilibrium of the Coulomb gas on this potential; the binding force of the potential balances the Coulomb repulsion of the rest of the particles. This produces some profile for the mean density of eigenvalues, with a vanishing denisty as one approaches the so-called spectral edges.

On the other hand, the Hamiltonian $H_{\mathrm{Haar}}$ has approximately uniform spectrum on an interval. Therefore, it can only arise as a highly atypical member of any ensemble of Hamiltonians defined by a smooth potential. In this sense, the Hamiltonian (1.3) is exponentially unlikely to be generated [17].

If, on the contrary, $H_{\mathrm{Haar}}$ is to represent a typical member of an ensemble of Hamiltonians, the measure in the space of Hamiltonians defining this ensemble has to be able to counter the eigenvalue repulsion locally to generate a uniform density close to the edges. This can only happen provided that the measure defined by the potential is extremely concentrated, like for instance for the case of an infinite well potential. Still, this is not enough to construct (1.3). Finer spectral properties, like the long-range eigenvalue repulsion, needs to be modified for $H_{\mathrm{Haar}}$ given that the phases of a Haar random matrix belong to the unitary circular ensemble [5]. For instance, eigenvalues at opposite edges of the interval must possess a strong repulsion, even if they are far away from each other. Therefore, the measure in the space of Hamiltonians needs to incorporate all of these features, which are highly artificial from the point of view of the universal features of random Hamiltonians.

Another way to see that the construction of $H_{\mathrm{Haar}}$ is extremely fine-tuned is to note that perturbing the time-evolution by a small amount of time would completely destroy the Haar-randomness of $U_{\mathrm{Haar}}$. In particular, at two units of time the ensemble of unitaries $U_{\mathrm{Haar}}^2 = \exp(-2iH_{\mathrm{Haar}})$ will fail to be distributed according to the Haar measure, and this will be very manifest even at the level of low moments of the distribution. In fact, if we only require to spoil exponentially high moments of the distribution, it will be enough to consider an exponentially small time-evolution $U_{\mathrm{Haar}}^t = \exp(-itH_{\mathrm{Haar}})$, for a time $|t - 1| \gtrsim O(\exp(-d \log d))$.[3] Such a dynamical instability is another piece of evidence that shows that this construction is extremely fine-tuned.

Our conjecture generalizes these considerations to the case of time-dependent Hamiltonians.

---

[3] This can be signaled e.g. from the $k$-th frame potential [18] which for the ensemble $U_{\mathrm{Haar}}^n$ scales as $n^k k!$.

**A robust linear growth in design for Brownian systems**

The dynamical approach to randomness can be quantified in quantum information theoretic terms by introducing a notion of $k$-randomness known as a *unitary k-design*: a distribution of unitaries which reproduces the first $k$ moments of the Haar distribution. Unitary designs play a prominent role in many corners of quantum information theory [19–31], many-body quantum chaos [18, 32–34] and in models of black holes [6].

In this paper, we will continue the study initiated in [33] and analyze how $k$-randomness is generated as a function of time for a general class of time-dependent Hamiltonians. The continuous systems that we will consider emulate random circuits in that they are defined by picking the couplings of the operators that define the Hamiltonian randomly and independently at each instant of time (see Fig. 1). This is, in some sense, the most random dynamical evolution one can consider, while being restricted to $p$-body interactions. Intuitively, such systems define a Brownian motion in $\mathsf{U}(d)$, which is inherited from the exponentiation of the standard Brownian motion in the algebra $\mathfrak{u}(d)$ of Hamiltonians. For $p$-local systems, the Brownian motion is restricted to the $p$-local directions of $\mathfrak{u}(d)$. For completely non-local systems, the Brownian motion is unrestricted.

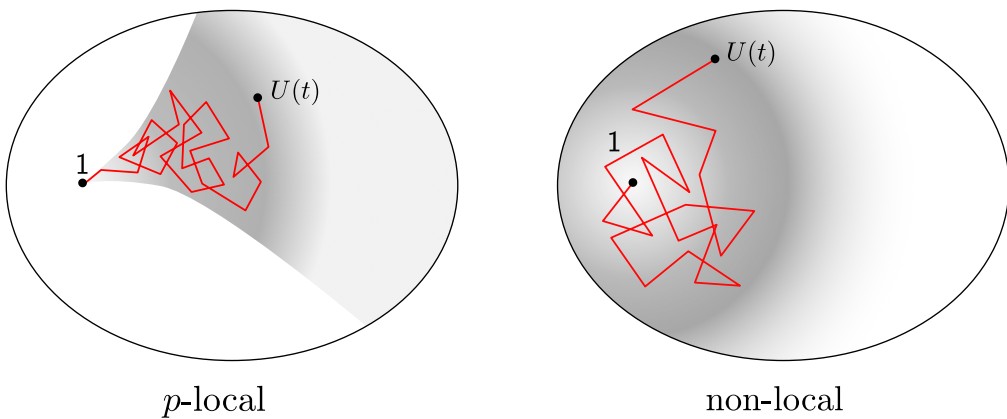

$p$-local             non-local

**Figure 1:** In the $p$-local Brownian model, the Hamiltonian at each time step is restricted to be few-body. In the completely non-local Brownian model the Hamiltonian is a random matrix at each time step and the Brownian motion in $\mathsf{U}(d)$ is unrestricted. The formation of an approximate $k$-design takes a time (up to logarithmic factors) linear in $k$ in both cases.

As we will show in this paper, both for the $p$-local and the non-local cases, the time to $k$-design for a Brownian system is linear in $k$. As a consequence, this makes the time to reach the Haar distribution exponential in the system size. Therefore, our results imply that such generic time-dependent Hamiltonians are slow generators of randomness, independently of the degree of locality of the Hamiltonian.

Note that, in actual random quantum circuits with few-body gates, the linear growth of the time to design with $k$ is known to be optimal, for the reason that discrete $k$-designs have $\exp(k)$ many distinct unitaries. The non-trivial part the becomes to indeed show that this linear growth is saturated. While it is known that $k$-designs with a polynomial value of $k$ are formed efficiently, the proof of general linear growth remains elusive [23–25, 28–30, 35–37]. Our results for $p$-local Brownian systems signal that the linear growth is generally saturated for the continuous version of these random circuits.

On the other hand, the non-local systems that we study in this paper involve continuous time evolutions with fairly general, and even completely non-local, time-dependent Hamiltonians. In this case, the analogy with random quantum circuit with arbitrary non-local unitary gates is no longer useful, and there is no obvious upper bound on the growth in design. The fact that we find such a slow growth of randomness for non-local Brownian models provides evidence in favor of our conjecture.

**Complexity vs randomness**

In light of our conjecture, the main goal of this paper is to make a quantitative distinction of the many-body quantum systems whose dynamical evolution can be catalogued as a 'hyperfast quantum computation' from the systems whose time-evolution corresponds to a 'hyperfast randomization'. The former are expected to be very common among the most general non-local systems; however, our conjecture suggest that the latter are extremely rare.

In this direction, invoking the bounds on "complexity by design" derived in [18] and used in [32], the robust linear growth in design for Brownian systems translates into a growth which is at least linear for the circuit complexity of the time-evolution operator $U(t)$ for these systems. These bounds also have an avatar for the strong notion of quantum complexity introduced in [38]. Strong complexity is more tightly related to the generation of randomness, in that it measures the operational cost of distinguishing the unitary channel associated to $U$ from a perfect randomizer (the maximally depolarizing channel).

On the other hand, we will argue that such estimations provide very loose lower bounds for the circuit complexity of $U(t)$, at the very least when the latter is drawn from a unitary design formed dynamically by an ensemble of non-local Hamiltonians.

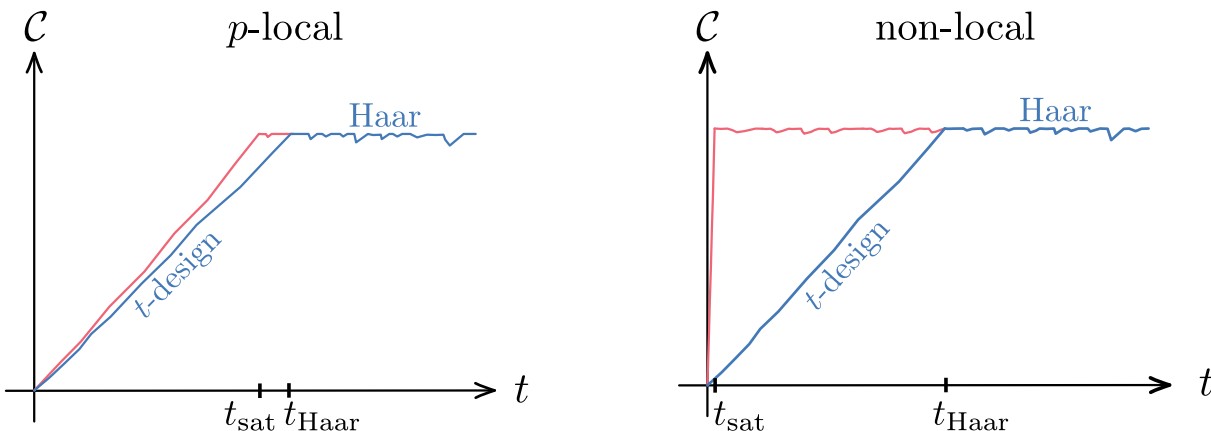

**Figure 2:** Comparison between the expected linear growth of circuit complexity (in red) and the linear growth of the lower bound in complexity for a design (in blue) that we compute for a typical Brownian $p$-body Hamiltonian. We observe that there is a parametric separation between the time at which circuit complexity saturates $t_{\text{sat}}$ and the time to reach approximate Haar random $t_{\text{Haar}}$. On the left, the situation for a $p$-local system, where both timescales are exponential in system size ($\log d$). On the right, the situation for a completely non-local system, where saturation of circuit complexity occurs at a poly($\log d$) timescale, while the time to Haar is poly($d$), exponential in system size.

In particular, as shown in Fig. 2, these bounds imply a robust linear growth on complexity for

both the $p$-local and the non-local case, with a slope proportional to $p$, which is another manifestation of the linear growth in desgin. On the other hand, the growth in circuit complexity is expected to generally be hyperfast in non-local systems. A heuristic argument for this is that, after Trotterization of the time-evolution operator at early times, a general non-local Hamiltonian is exponentially complex to implement with any universal set of few-body gates (see e.g. [39]). These different behaviors simply represent that

$$U \text{ is maximally complex } \implies U \text{ is Haar random}, \tag{1.4}$$

and thus $U(t)$ can be maximally complex at relatively early times, but it does not become Haar random in the space of unitaries until exponentially late times in the system size.

The paper is organized as follows. In section 2 we will include background material to make the paper self-contained. We will define the notion of approximate unitary design, and derive a way to compute the trace distance to design for general Brownian systems. We will also state a sufficient set of assumptions which guarantee a linear growth in design. We will finally relate all of these findings to the spectral properties of the time-evolution operator. In section 3 we will apply these considerations to the Brownian $p$-SYK model as a function of $p$. We will study the transition between $p$-local and non-local regimes in the double-scaling limit. In section 4 we will propose a completely non-local Brownian random matrix model for the Hamiltonian and study the dynamical generation of randomness in this model, making contact with the non-local limit of the $p$-SYK model. For the Brownian GUE model, we will reformulate our results in terms of classical Brownian motion in $\mathsf{U}(d)$. In section 5 we will compare the slow generation of randomness with the expected hyperfast growth of circuit complexity for non-local systems. We will provide an alternative argument, using complexity geometry, of the expected transition between linear to hyperfast complexity growth for a Brownian system as the degree of locality is scaled. We will end with conclusions and leave complementary material for the appendices.

## 2  Generalities: formation of approximate designs

Let us consider a general Hilbert space $\mathcal{H}$ of dimension $d$. We will study disordered and possibly time-dependent Hamiltonians $H(t)$ which generate an ensemble $\mathcal{E}_H$, formally defined by some measure $\mathcal{D}H(t)$ in the space of time-dependent Hamiltonians. From this ensemble we can form, at a fixed time $t$, the corresponding ensemble of time-evolution unitaries

$$\mathcal{E}_t = \left\{ U(t) = \mathsf{T} \exp\left( -i \int_0^t \mathrm{d}s H(s) \right) : H(t) \in \mathcal{E}_H \right\}. \tag{2.1}$$

In general, given any ensemble $\mathcal{E}$ of unitary operators, its $k$-th moment map $\Phi_{\mathcal{E}}^{(k)}$ is defined as the quantum channel

$$\Phi_{\mathcal{E}}^{(k)}(X) \equiv \mathbb{E}\left[ U^{\otimes k} X (U^\dagger)^{\otimes k} \right], \tag{2.2}$$

acting on operators $X \in \mathrm{End}(\mathcal{H}^{\otimes k})$. The quantum channel $\Phi_{\mathcal{E}}^{(k)}$ can equivalently be represented as the

superoperator

$$\hat{\Phi}_{\mathcal{E}}^{(k)} \equiv \mathbb{E}\left[U(t)^{\otimes k} \otimes U(t)^{*\,\otimes k}\right],\tag{2.3}$$

acting on the replicated Hilbert space $\mathcal{H}^{\otimes 2k}$. We will mostly use the superoperator representations of $\Phi_{\mathcal{E}}^{(k)}$ in this paper. If $\mathcal{E}^{\dagger} = \mathcal{E}$, then the superoperator is Hermitian, $(\hat{\Phi}_{\mathcal{E}}^{(k)})^{\dagger} = \hat{\Phi}_{\mathcal{E}}^{(k)}$.

A unitary ensemble $\mathcal{E}$ is a *k-design* if its $k$-th moment map $\Phi_{\mathcal{E}}^{(k)}$ agrees with the $k$-th moment map of the Haar ensemble,[4]

$$\mathcal{E} \text{ is a } k\text{-design} \quad \Leftrightarrow \quad \hat{\Phi}_{\mathcal{E}}^{(k)} = \hat{\Phi}_{\text{Haar}}^{(k)}.\tag{2.4}$$

More generally, we can define an *$\varepsilon$-approximate k-design* as a unitary ensemble $\mathcal{E}$ which is effectively indistinguishable from an exact $k$-design up to some tolerance $\varepsilon$. For our purposes, the relevant notion of distinguishability between quantum channels is provided by the trace distance between the corresponding superoperators. Therefore, we define

$$\mathcal{E} \text{ is an } \varepsilon\text{-approximate } k\text{-design} \quad \Leftrightarrow \quad \left\|\hat{\Phi}_{\mathcal{E}_t}^{(k)} - \hat{\Phi}_{\text{Haar}}^{(k)}\right\|_1 < \varepsilon.\tag{2.5}$$

Note that alternative definitions of approximate designs exist, most commonly in terms of the diamond distance between quantum channels [15, 40]. These definitions are all related by factors of the dimension (see e.g. [41]); we recall the relation between (2.5) and the diamond definition of design in appendix B.

As a general remark, it is important to realize that many ensembles $\mathcal{E}_t$ fail to form approximate unitary $k$-designs even for small $k$ at arbitrary late times, and thus a time-evolution operator $U(t) \in \mathcal{E}_t$ drawn randomly from these ensembles does never become Haar typical. An interesting example is that of an ensemble of unitaries $\mathcal{E}_t$ generated dynamically by time-independent Hamiltonians $H$ in some universality class of random matrices. Such systems were studied extensively in [32]. The time-evolution operator $U(t)$ in these cases is far from Haar random even at the timescale set by the inverse mean level spacing of the Hamiltonian, $t_{\Delta E} \sim \Delta E^{-1}$. The discrepancy between $U(t_{\Delta E})$ and a Haar random unitary is manifest at the level of the spectral properties of $U(t_{\Delta E})$ (this argument has been presented in [18, 32]). The spectrum of $U(t_{\Delta E})$ consists, with extremely high probability, of statistically uncorrelated phases whose phase-separations are Poisson distributed. On the contrary, the spectrum of a Haar random unitary consists of correlated phases that tend to repel each other. These cases trivially satisfy our conjecture, given that they are not able to generate randomness even in the long run.[5]

---

[4] In appendix A we recall some basic facts about $\Phi_{\text{Haar}}^{(k)}$.

[5] In [32] it was argued that the ensemble $\mathcal{E}_t$ inherited from the GUE of Hamiltonians becomes an approximate $k$-design at the dip time $t_d \sim \sqrt{d}$, where the spectral correlations of the Hamiltonian are suppressed, for values $k \ll d$. This system would still satisfy our conjecture given that the (finite-temperature) dip timescale is exponential in the system size; however, it is unclear whether the time-evolution operators ever becomes approximate Haar random in this system, since this requires to capture an exponentially large number of moments of the Haar distribution $k \gtrsim d$.

**A bound on the trace distance from the frame potential**

In previous work [18, 28, 32, 33], the strategy to estimate the trace or diamond distance between the $k$-th moment maps was to define the *k-th frame potential* of the unitary ensemble $\mathcal{E}$,

$$F_{\mathcal{E}}^{(k)} \equiv \mathbb{E}\left[ \left| \mathrm{Tr}(V^\dagger U) \right|^{2k} \right], \tag{2.6}$$

where the expectation value is understood with respect to both $U, V \in \mathcal{E}$ independently. In practice, the frame potential is manifestly simpler to compute than the trace or diamond distance. Moreover, the frame potential is related to measures of early-time chaos; in particular, it captures an average out of time-order correlator [18].

The difference in frame potentials $F_{\mathcal{E}}^{(k)} - F_{\mathrm{Haar}}^{(k)}$ is the square of the 2-norm distance between the $k$-th moment superoperators

$$F_{\mathcal{E}}^{(k)} - F_{\mathrm{Haar}}^{(k)} = \left\| \hat{\Phi}_{\mathcal{E}}^{(k)} - \hat{\Phi}_{\mathrm{Haar}}^{(k)} \right\|_2^2 \geqslant 0. \tag{2.7}$$

where we recall that the subscript 2 simply represents the standard Schatten 2-norm $\|A\|_2 = \sqrt{\mathrm{Tr}(AA^\dagger)}$, to be distinguished from the 2-norm distance between quantum channels (see e.g. [41] for the different definitions). The frame potential for the Haar ensemble is $F_{\mathrm{Haar}}^{(k)} = k!$.

By Hölder's inequality it follows that

$$\left\| \hat{\Phi}_{\mathcal{E}}^{(k)} - \hat{\Phi}_{\mathrm{Haar}}^{(k)} \right\|_1^2 \leqslant d^{2k} \left( F_{\mathcal{E}}^{(k)} - F_{\mathrm{Haar}}^{(k)} \right), \tag{2.8}$$

An analogous bound for the diamond distance follows from (B.2) (see also [28]). Therefore, the frame potential can be used as a sufficient condition for the formation of an approximate design in real time

$$d^k \left( F_{\mathcal{E}_t}^{(k)} - F_{\mathrm{Haar}}^{(k)} \right)^{1/2} < \varepsilon \quad \Rightarrow \quad \mathcal{E}_t \text{ is an } \varepsilon\text{-approximate } k\text{-design}. \tag{2.9}$$

However, the factor of the dimension in (2.8) indicates that, unless the frame potential is close to the Haar value, $F_{\mathcal{E}_t}^{(k)} - F_{\mathrm{Haar}}^{(k)} < O(1)$, the upper bound cannot be tight, given that the trace distance is at most $d^k$ (note that $F_{\mathcal{E}_0}^{(k)} - F_{\mathrm{Haar}}^{(k)} = d^{2k} - k!$). With additional information about the channels, we will now provide a way to compute the trace distance to a $k$-design for general Brownian systems.

## 2.1 Moment maps for Brownian systems

In order to proceed, we shall first explain some general features of the quantum channels $\Phi_{\mathcal{E}}^{(k)}$ associated to Brownian systems. Consider a general time-dependent Hamiltonian

$$H(t) = \sum_{\alpha=1}^{K} J_\alpha(t)\, \mathcal{O}_\alpha. \tag{2.10}$$

where the $\mathcal{O}_\alpha$ are a set of $K$ Hermitian operators, normalized to satisfy $\mathrm{Tr}(\mathcal{O}_\alpha \mathcal{O}_\alpha^\dagger) = d$. The Brownian ensemble is defined by the specification of the disordered time-dependent couplings $J_\alpha(t)$, formally given in terms of the path integral measure

$$\mathbb{E}[\bullet] = \frac{1}{\mathcal{N}} \int \prod_\alpha \mathcal{D}J_\alpha(t) \, e^{-\frac{K}{2J} \int \mathrm{d}t \sum_\alpha J_\alpha(t)^2} (\bullet) . \tag{2.11}$$

The normalization constant $\mathcal{N}$ is determined by the condition $\mathbb{E}[1] = 1$. This ensemble produces independent gaussian white-noise correlated random variables, with zero mean and variance

$$\mathbb{E}\left[J_\alpha(t)J_{\alpha'}(0)\right] = \frac{J}{K}\delta(t)\delta_{\alpha\alpha'} , \tag{2.12}$$

where $J$ has dimensions of energy. With this normalization, the Hamiltonian has, at each instant of time, zero expectation value and fixed variance in the maximally mixed state $\rho_0 = \frac{1}{d}$,

$$\mathbb{E}\left[\mathrm{Tr}(\rho_0 H(t))\right] = 0 , \tag{2.13}$$

$$\mathbb{E}\left[\mathrm{Tr}(\rho_0 H(t)H(0))\right] = J\delta(t) . \tag{2.14}$$

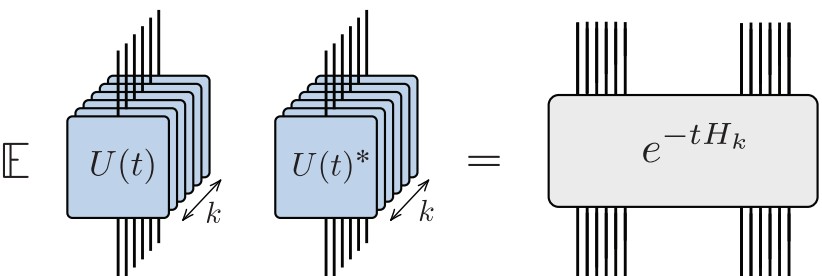

**Figure 3:** The $k$-th moment superoperator $\hat{\Phi}_{\mathcal{E}_t}^{(k)}$ is the unnormalized thermal state for the effective Hamiltonian $H_k$ at inverse temperature $t$. The Hamiltonian $H_k$ is time-independent and couples the $2k$ replicas together.

Under such assumptions, the time-dependent problem of analyzing the moment superoperator $\hat{\Phi}_{\mathcal{E}_t}^{(k)}$ reduces to an effective equilibrium thermodynamic problem for an interacting Hamiltonian between replicas, at inverse temperature $t$, as illustrated in Fig. 3. More precisely, the $k$-th moment superoperator $\hat{\Phi}_{\mathcal{E}_t}^{(k)}$ can be explicitly rewritten as the unnormalized thermal density matrix

$$\hat{\Phi}_{\mathcal{E}_t}^{(k)} = \frac{1}{\mathcal{N}} \int \prod_\alpha \mathcal{D}J_\alpha \, e^{-\frac{K}{2J} \int \mathrm{d}t \sum_\alpha J_\alpha(t)^2} U(t) \otimes ... \otimes U(t)^* = \exp\left(-tH_k\right) , \tag{2.15}$$

for the time-independent effective Hamiltonian [33]

$$H_k = \frac{J}{2K} \sum_{\alpha=1}^K \left( \sum_{r=1}^k \mathcal{O}_\alpha^r - \mathcal{O}_\alpha^{\bar{r} \, *} \right)^2 . \tag{2.16}$$

In this notation $r = 1,...,k$ labels the 'forward replica' in which the operator $\mathcal{O}_\alpha^r$ acts. The name

forward replica simply comes from the orientation of the time-evolution; these replicas are associated to the $r$-th $U$ factor of the moment map. On the contrary $\bar{r} = 1, ..., k$ labels the 'backward replica' in which $\mathcal{O}_\alpha^{\bar{r}}$ acts, and which is associated to the $r$-th $U^*$ factor. Due to the square in (2.16), the effective Hamiltonian $H_k$ contains bi-local couplings between replicas.

**Late time limit from the ground space**

Let $\mathrm{GS}_k$ denote the ground space of the effective Hamiltonian $H_k$ defined in (2.16). At infinite time, or zero effective temperature, the $k$-th moment superoperator becomes the orthogonal projector into this subspace,

$$\hat{\Phi}_{\mathcal{E}_\infty}^{(k)} = \exp\left(-\infty H_k\right) = \Pi_{\mathrm{GS}_k}\,. \tag{2.17}$$

Therefore, by constructing $\mathrm{GS}_k$ one learns about the expected late-time limiting behavior of the quantum channel $\Phi_{\mathcal{E}_t}^{(k)}$ for a Brownian system.

In general, the effective Hamiltonian (2.16) contains the following set of ground states,

$$|W_\sigma\rangle = \bigotimes_{r=1}^k |\infty\rangle_{r\sigma(\bar{r})}\,, \qquad H_k |W_\sigma\rangle = 0\,, \tag{2.18}$$

where $\sigma \in \mathrm{Sym}(k)$. To explain our notation, the state $|W_\sigma\rangle$ simply corresponds to the product of $k$ infinite-temperature thermofield double (TFD) states $|\infty\rangle_{r\bar{s}} = \frac{1}{\sqrt{d}} \sum_{i=1}^d |i, i\rangle_{r\bar{s}}$, where $\{|i\rangle\}_{i=1}^d$ is some arbitrary orhonormal basis of $\mathcal{H}$. The permutation $\sigma$ labels a given pairing of the $k$ forward and $k$ backward replicas, given that $r$ and $\sigma(\bar{r})$ are paired in the TFD. The $k!$ states $\{|W_\sigma\rangle : \sigma \in \mathrm{Sym}(k)\}$ are not exactly mutually orthogonal, but they are linearly independent as long as $k \leqslant d$ (see e.g. [42–44]).

Let $V_k$ be the subspace of the ground space generated by this set of ground states,

$$V_k = \mathrm{Span}\{|W_\sigma\rangle : \sigma \in \mathrm{Sym}(k)\} \subset \mathrm{GS}_k\,. \tag{2.19}$$

The superoperator $\hat{\Phi}_{\mathrm{Haar}}^{(k)}$ is the orthogonal projector into this subspace

$$\hat{\Phi}_{\mathrm{Haar}}^{(k)} = \Pi_{V_k} = \sum_{\sigma, \tau \in \mathrm{Sym}(k)} (Q^{-1})_{\sigma\tau} |W_\sigma\rangle\langle W_\tau|\,, \tag{2.20}$$

where $Q_{\sigma\tau} = \langle W_\sigma | W_\tau \rangle$ and $k \leqslant d$. We will recall why this is true in appendix A.

Therefore, we see that the necessary and sufficient condition for the Brownian ensemble to become an exact $k$-design at infinite times is

$$\Phi_{\mathcal{E}_\infty}^{(k)} = \Phi_{\mathrm{Haar}}^{(k)} \quad \Leftrightarrow \quad V_k = \mathrm{GS}_k\,, \tag{2.21}$$

In practice, one only needs to check this condition for $k \lesssim d$, since approximate $k$-designs are also approximate $(k + 1)$-designs for $k \gtrsim d$ [18].

The condition (2.21) will be met as long as the effective Hamiltonian has no other linearly independent ground states to the states of the form (2.18). Generally, if the set of operators $\mathcal{O}_\alpha$ is all-to-all, in the sense that there are no clusters among the degrees of freedom, and in the absence of global symmetries for the ensemble, this is expected to be true; however, proving this requires of the particular form of the operators $\mathcal{O}_\alpha$ in (2.16). In appendix C we show that as a counterexample, the conservation of energy in a Brownian system leads to a situation in which randomness is never generated.

## 2.2   Trace distance as a thermal partition function

Given (2.15), in the case of Brownian systems, it becomes straightforward to compute the Schatten $q$-norm distance between the $k$-th moment superoperators $\hat{\Phi}_{\mathcal{E}_t}^{(k)} = \exp(-tH_k)$ and $\hat{\Phi}_{\mathcal{E}_{\mathrm{Haar}}}^{(k)} = \Pi_{V_k}$, given that these superoperators can be simultaneously diagonalized. Additionaly, the spectrum of $\hat{\Phi}_{\mathcal{E}_t}^{(k)}$ is real and non-negative, while $\hat{\Phi}_{\mathcal{E}_{\mathrm{Haar}}}^{(k)}$ vanishes in the orthogonal subspace to $V_k$.

To this end, we will define the effective thermal partition function at inverse temperature $t$,

$$Z_k(t) \equiv \mathrm{Tr}\left(\exp\left(-tH_k\right)\right). \tag{2.22}$$

In terms of this partition function, the $q$-norm distance simply reads

$$\left\|\hat{\Phi}_{\mathcal{E}_t}^{(k)} - \hat{\Phi}_{\mathrm{Haar}}^{(k)}\right\|_q = \left(Z_k(qt) - \dim(V_k)\right)^{\frac{1}{q}}, \qquad \text{where } \dim(V_k) = k!. \tag{2.23}$$

Note that in this form the difference of frame potentials corresponds to $F_{\mathcal{E}_t}^{(k)} - F_{\mathrm{Haar}}^{(k)} = Z_k(2t) - k!$. The properties (2.20) and (2.21) directly follow from (2.23); the $q$-norm distance between both superoperoperators vanishes at infinite late times if and only if $Z_k(\infty) = \dim(\mathrm{GS}_k) = k!$, which implies that $V_k = \mathrm{GS}_k$, given that $V_k$ is always a subspace of $\mathrm{GS}_k$.

This formula also shows that the $q$-norm distance (2.23) to design is non-increasing as a function of time

$$\frac{\mathrm{d}}{\mathrm{d}t}\left\|\Phi_{\mathcal{E}_t}^{(k)} - \Phi_{\mathrm{Haar}}^{(k)}\right\|_q = -Z(qt)^{\frac{1}{q}}\left(1 - \frac{k!}{Z(qt)}\right)^{\frac{1-q}{q}}\langle H_k\rangle_{qt} \leqslant 0, \tag{2.24}$$

where $\langle H_k\rangle_\beta = Z_k(\beta)^{-1}\mathrm{Tr}\left(H_k\exp\left(-\beta H_k\right)\right)$ is the effective energy at inverse temperature $\beta$. This quantity is possitive by virtue of $H_k \geqslant 0$. This feature is expected from the Markovian nature of the Brownian systems.

Using (2.23) for $q = 1$, we can reformulate the condition (2.5) for the formation of designs in Brownian systems

$$\mathcal{E} \text{ is an } \varepsilon\text{-approximate } k\text{-design} \quad \Leftrightarrow \quad Z_k(t) - k! < \varepsilon. \tag{2.25}$$

Given these considerations, the *time to k-design* is defined as

$$t_k \equiv \inf_t \left\{t : Z_k(t) - k! < \varepsilon\right\}. \tag{2.26}$$

The goal of the rest of the paper is to compute $t_k$ for different Brownian systems, by analyzing the effective thermal partition function $Z_k(t)$ as a function of the degree of locality of the Hamiltonian in those systems. Before proceeding to specific examples, we shall make some general remarks about the expected behavior of $t_k$ for large values of $k$ under a certain set of general assumptions.

## 2.3 Linear time to design

We will now state a sufficient set of assumptions which guarantee a time to $k$-design $t_k$ proportional to $k$ for general Brownian systems. These assumptions, which will need to be verified for the different particular Brownian systems that we will study, are:

1. The ground space of the effective Hamiltonian $H_k$ (2.16) is linearly generated by the states of the form (2.18), that is,

$$\text{GS}_k = V_k \,. \tag{2.27}$$

As explained in section 2.1 this is equivalent to the assumption that $\mathcal{E}_\infty$ is a $k$-design.

2. The *spectral gap* $\Delta_k$ of the effective Hamiltonian $H_k$ is, for sufficiently large $k$, asymptotically independent of $k$,

$$\Delta_k \sim E_0 = O(k^0) \,. \tag{2.28}$$

3. The *number of first excited states* $N_*$ of the effective Hamiltonian $H_k$ scales exponentially with $k$,

$$N_* \sim \alpha^k \,. \tag{2.29}$$

The intuition behind 2 and 3 is that in systems with large number of degrees of freedom, each ground state of the effective Hamiltonian typically admits a mean field description in terms of $k$ semiclassical saddle points connecting each forward replica with a backward replica [33]. The first excited states then correspond to 'single particle excitations' on top of these mean field saddle points. As isolated excitations, their energy $\Delta_k$ should be independent of the number of saddlepoints there are, which is controlled by $k$. Moreover, the multiplicity $N_*$ of single-particle excitations should be proportional to the number of ground states, which scales exponentially with $k$.[6]

Assuming condition 1, at sufficiently late times, or small effective temperatures below the gap, the trace distance to design behaves asymptotically as

$$\left\| \hat{\Phi}^{(k)}_{\mathcal{E}_t} - \hat{\Phi}^{(k)}_{\text{Haar}} \right\|_1 = Z_k(t) - Z_k(\infty) \sim N_* e^{-t\Delta_k} \,, \tag{2.30}$$

where $N_*$ is the number of first excited states. Therefore, under the assumptions 2 and 3, this leads to

---

[6] In sparse systems, verifying assumptions 2 and 3 might be much more subtle. For instance, in spatially local random circuits, domain walls connecting different vacua can proliferate the IR of the effective Hamiltonian [28].

a time to $k$-design

$$t_k \sim \frac{1}{E_0} \left( k \log \alpha + \log \varepsilon^{-1} \right) , \tag{2.31}$$

which, up to logarithmic factors ($\alpha$ could depend on $k$), gives the promised linear time to design.

## 2.4 Randomness from the spectrum of $U(t)$

Another interesting observation is that, for Brownian systems, the trace distance to design only depends on the spectral properties of the time-evolution operator $U(t)$. To see this, we can use (2.15) and (2.22) to rewrite the effective thermal partition function as

$$Z_k(t) = \mathbb{E}\left[ \mathrm{SFF}(U(t))^k \right] , \tag{2.32}$$

where for any unitary $U$ we have introduced its unnormalized *spectral form factor* (SFF)

$$\mathrm{SFF}(U) \equiv |\mathrm{Tr}\, U|^2 . \tag{2.33}$$

Let us denote the eigenvalues of $U$ by $e^{i\theta_j}$ for $-\pi \leqslant \theta_j < \pi$ ($j = 1, ..., d$) and by $\rho(\theta) = \sum_j \delta(\theta - \theta_j)$ its eigenphase density. In terms of the eigenphases, the SFF reads

$$\mathrm{SFF}(U) = d + 2 \sum_{i<j} \cos(\theta_i - \theta_j) . \tag{2.34}$$

That is, the trace distance of $\mathcal{E}_t$ to a 1-design is controlled by the average value of $\cos(\theta_i - \theta_j)$ over the spectra of $U(t) \in \mathcal{E}_t$. Note that for Poisson distributed phase separations, and a uniform mean density of states, the average value of (2.34) is $d$. This is precisely what happens for systems with conserved energy, at timescales set by the inverse mean level spacing (i.e. at the *plateau* of the SFF [45]); we study an example of energy-conserving Brownian system in Appendix C.

On the other hand, for the Haar ensemble $\mathcal{E}_{\mathrm{Haar}}$, the average of (2.34) attains a much smaller value. This happens because of the stronger spectral correlations of the eigenvalues of a Haar random matrix; in particular, the phases tend to repel each other. The eigenphases of a Haar random matrix are distributed according to the circular unitary ensemble, with joint probability distribution [5]

$$p_{\mathrm{Haar}}(\theta_1, ..., \theta_d) = \frac{1}{(2\pi)^d d!} \prod_{i<j} \left| e^{i\theta_i} - e^{i\theta_j} \right|^2 . \tag{2.35}$$

This probability distribution produces a uniform mean eigenphase density $\mathbb{E}\left[\rho(\theta)\right] = \frac{d}{2\pi}$ and a connected two-point correlation for $\delta\rho(\theta) = \rho(\theta) - \mathbb{E}[\rho(\theta)]$,

$$\mathbb{E}\left[\delta\rho(\theta)\delta\rho(\theta')\right] = -\frac{\sin\left(d\,\frac{\theta-\theta'}{2}\right)^2}{\left(2\pi \sin\left(\frac{\theta-\theta'}{2}\right)\right)^2} , \tag{2.36}$$

which reduces to the usual sine kernel for a Gaussian random matrix in the regime $|\theta - \theta'| \ll \pi$.

In particular, the eigenvalue repulsion between phases is responsible of lowering the average value of the SFF. In the large-$d$ limit we have

$$\frac{1}{d^2} \int_{-\pi}^{\pi} \mathrm{d}\theta \int_{-\pi}^{\pi} \mathrm{d}\theta' \, \mathbb{E}\left[\rho(\theta)\rho(\theta')\right] \cos\left(\theta - \theta'\right) = -\frac{1}{2d}\,. \tag{2.37}$$

This allows to take the expectation value of (2.34) directly from the spectral properties of Haar random matrices, and this way obtain $\mathbb{E}\left[\mathrm{SFF}(U)\right] = 1$. Note that, even if the eigenvalue repulsion produces an $O(d^{-1})$ effect individual terms in the sum (2.34), the fact that the total sum contains $O(d^2)$ terms makes it relevant for the expectation value of the SFF.

Similarly, for $k \leqslant d$, the average value of the $k$-th power of the SFF over the Haar ensemble is given by $\mathbb{E}\left[\mathrm{SFF}(U)^k\right] = k!$, which is exponentially smaller in the entropy than the value $d^k k!$ for Poisson distributed phases. This value arises from the $2k$-th moments of the spectral density, $\mathbb{E}[\rho(\theta_{i_1})...\rho(\theta_{i_{2k}})]$. Using the inversion formula, we can directly read the induced probability distribution $p_{\mathrm{Haar}}(\mathrm{SFF})$ associated to these moments,[7]

$$p_{\mathrm{Haar}}(\mathrm{SFF}) \approx \exp(-\mathrm{SFF})\,. \tag{2.38}$$

For Brownian systems, from the general considerations of section 2.1, the eigenvalue repulsion in the spectrum of $U(t)$ will be dynamically generated as a function of time. Given that (2.32) is non-increasing and $Z_k(t) \geqslant k!$, assuming that the system satisfies the assumptions of section 2.3, the joint probability distribution for the eigenvalues of $U(t)$ will converge weakly to (2.35) at infinite times. This will be signaled by the induced probability distribution $p_t(\mathrm{SFF})$ for the SFF from ensemble of unitaries $\mathcal{E}_t$, which will converge weakly to (2.38). As an example, for $k = 1$ the two-point eigenvalue repulsion that $U(t)$ will develop will be responsible of the plateau value of the SFF for a Brownian system. At finite times, the ensemble $\mathcal{E}_t$ becomes an approximate $k$-design when the $2k$-th spectral moment $\mathbb{E}[\rho(\theta_{i_1})...\rho(\theta_{i_{2k}})]$ approximately agrees with that of (2.35). This will take a time which is parametrically linear in $k$ for Brownian systems satisfying the assumptions of section 2.3.

## 3    Brownian SYK: from $p$-local to non-local

In this section, we will study the generation of randomness in the Brownian $p$-SYK model, a system of $N$ Majorana fermions $\psi_1, ..., \psi_N$ satisfying the anticommutation relations $\{\psi_i, \psi_j\} = 2\delta_{ij}$. The Hamiltonian of the system is all-to-all and couples the fermions in groups of even $p$,

$$H(t) = i^{\frac{p}{2}} \sum_{i_1 < ... < i_p} J_{i_1...i_p}(t) \, \psi_{i_1}...\psi_{i_p}\,. \tag{3.1}$$

---

[7] This distribution is related to the Porter-Thomas distribution, given that $\mathrm{Tr}(U)$ can be interpreted as the sum over survival amplitudes in any basis of states.

The Brownian model is defined by (2.11) for the independent random couplings $J_{i_1...i_p}(t)$, with $K = \binom{N}{p}$. That is, the couplings possess gaussian white-noise correlations, with zero mean and a two-point function

$$\mathbb{E}\left[J_{i_1...i_p}(t)J_{i'_1...i'_p}(0)\right] = \frac{J}{\binom{N}{p}}\delta(t)\delta_{i_1,i'_1}...\delta_{i_p,i'_p}. \tag{3.2}$$

With this normalization of the couplings, the Hamiltonian at each instant of time has zero mean and fixed variance in the maximally mixed state $\rho_0 = \frac{1}{d}$,

$$\mathbb{E}\left[\mathrm{Tr}\left(\rho_0 H(t)H(0)\right)\right] = J\delta(t). \tag{3.3}$$

Note that our choice of normalization of the couplings is different from the normalization of e.g. [46] by a factor of $N/p^2$ (commonly denoted by $2/\lambda$ in the double-scaled SYK literature). The reason for our choice of normalization is that it is convenient to select a $p$-independent Brownian amplitude (3.3), so that we can properly compare how randomness is generated for different values of $p$. Moreover, we will make the amplitude $N$-independent to make our choice consistent with (2.14); however, restoring the factors of $N$ is straightforward.

As we derived in section 2.1 from the structure of the Hamiltonian and the random couplings, the $k$-th moment superoperator $\hat{\Phi}_{\mathcal{E}_t}^{(k)}$ can be written as the unnormalized thermal state (2.15) for the effective Hamiltonian

$$H_k = \frac{J}{2\binom{N}{p}}i^p\sum_{i_1<...<i_p}\left(\sum_{r,\bar{r}=1}^{k}\left[\psi_{I_p}^r - (-1)^{\frac{p}{2}}\psi_{I_p}^{\bar{r}}\right]\right)^2, \tag{3.4}$$

where we used the shorthand notation $\psi_{I_p} = \psi_{i_1}...\psi_{i_p}$ for the collective fermion variables. Again, the index $r(\bar{r})$ labels the forward (backward) replica. The factor of $(-1)^{\frac{p}{2}}$ represents the fact that for $p \equiv 2 \pmod 4$ the system does not possess time-reflection symmetry, and the complex conjugate of the operators in the backward replicas pick up a relative minus sign.

**Global symmetries: fermion parity**

We shall consider $N$ to be even in order to simplify the discussion. Let us recall some basic facts about the model. The Hilbert space $\mathcal{H}$, of dimension $d = 2^{\frac{N}{2}}$, consists of $N/2$ Dirac fermions [45, 46]

$$c_j = \frac{\psi_j - i\psi_{N-j}}{\sqrt{2}}, \qquad c_j^\dagger = \frac{\psi_j + i\psi_{N-j}}{\sqrt{2}} \qquad j = 1,...,\frac{N}{2}. \tag{3.5}$$

For our purposes, it will be important to consider the fermion number operator

$$Q = \sum_{i=1}^{N/2}c_i^\dagger c_i. \tag{3.6}$$

In particular, fermion parity $G = (-1)^Q$ is conserved by each Hamiltonian of the form (3.1). This implies that $[G, H_k] = 0$ at the level of the effective Hamiltonian. Due to this symmetry, there will

be many more ground states of the effective Hamiltonian $H_k$ (3.4) and $V_k$ will in this case be a proper subspace of $\mathrm{GS}_k$. The superoperators $\hat{\Phi}^{(k)}_{\mathcal{E}_t}$ and $\hat{\Phi}^{(k)}_{\mathrm{Haar}}$ will never be close in trace distance, and the ensemble $\mathcal{E}_t$ will never become an approximate $k$-design for any $k$.

To circumvent this complication that arises only due to the fermionic nature of the model, we can study the generation of randomness on each superselection sector of $G$. That is, we consider the decomposition $\mathcal{H} = \mathcal{H}^+ \oplus \mathcal{H}^-$ in odd an even fermion parity sectors, and ensembles $\mathcal{E}_U$ of unitaries $U$ of block diagonal form in this decomposition, satisfying $[G, U] = 0$. There will a block-diagonal Haar ensemble $\mathcal{E}_{\mathrm{Haar}_G}$ (see appendix A). In what follows, we will study the generation of block-diagonal randomness of $\mathcal{E}_t$ for the Brownian SYK model.

To do that, let us come back to the effective Hamiltonian (3.4) and note that there are now two linearly independent infinite-temperature TFD states between replicas $r$ and $\bar{s}$, due to the two fermion parity sectors. These are defined to be the Fock vacuum state and the maximally excited state of the $N$ $r\bar{s}$-Dirac fermions,

$$
\left.
\begin{aligned}
\left(\psi_j^r + i\,\psi_j^{\bar{s}}\right)|\infty, 1\rangle_{r\bar{s}} &= 0 \\[2mm]
\left(\psi_j^r - i\,\psi_j^{\bar{s}}\right)|\infty, 2\rangle_{r\bar{s}} &= 0
\end{aligned}
\right\} \quad \forall\, j = 1, ..., N\,.
\tag{3.7}
$$

The two states are related by fermion parity, $|\infty, 1\rangle_{r\bar{s}} = (-1)^{Q_r} |\infty, 2\rangle_{r\bar{s}}$. In particular, we can define fermion even and odd infinite-temperature TFD states

$$
|\infty, \pm\rangle_{r\bar{s}} = \frac{1 \pm (-1)^{Q_r}}{2} |\infty, 1\rangle_{r\bar{s}}\,.
\tag{3.8}
$$

These states are also even and odd with respect to $(-1)^{Q_{\bar{s}}}$.

The ground space of the effective Hamiltonian (3.4) is linearly generated by states of the form [33]

$$
|W_\sigma^v\rangle = \bigotimes_{r=1}^{k} |\infty, v_r\rangle_{r\sigma(\bar{r})}\,, \qquad H_k |W_\sigma^v\rangle = 0\,,
\tag{3.9}
$$

where $\sigma \in \mathrm{Sym}(k)$ and $v \in \{\pm\}^k$. These are simply the product of $k$ infinite-temperature TFD states between forward and backward replicas, paired by the permutation $\sigma$, with a suitable choice of fermion parity for each TFD, represented by $v$. That is, we find

$$
\mathrm{GS}_k = \mathrm{Span}\{|W_\sigma^v\rangle : \sigma \in \mathrm{Sym}(k), v \in \{\pm\}^k\} \equiv V_k^G\,.
\tag{3.10}
$$

By the considerations in appendix A, the corresponding superoperator $\hat{\Phi}^{(k)}_{\mathrm{Haar}_G}$ for the block-diagonal Haar ensemble is the orthogonal projector into this subspace,

$$
\hat{\Phi}^{(k)}_{\mathrm{Haar}_G} = \Pi_{V_k^G} = \sum_{\substack{\sigma, \tau \in \mathrm{Sym}(k) \\ v, w}} (Q^{-1})^{vw}_{\sigma\tau} |W_\sigma^v\rangle\langle W_\tau^w|\,,
\tag{3.11}
$$

where $Q^{vw}_{\sigma\tau} = \langle W_\tau^v | W_\sigma^w \rangle$ and $k \leqslant d$.

The generation of an $\varepsilon$-approximate block diagonal $k$-design can be then diagnosed by the trace distance to the block-diagonal Haar moment superoperators,

$$\left\|\hat{\Phi}_{\mathcal{E}_t}^{(k)} - \hat{\Phi}_{\mathrm{Haar}_G}^{(k)}\right\|_1 = Z_k(t) - Z_k(\infty), \tag{3.12}$$

where $Z_k(\infty) = \dim(\mathrm{GS}_k) = 2^k k!$.

**Particle-hole symmetry**

Let us finally comment on additional global symmetries, which arise as a function of the value of $N$ (mod 8) and can affect the discussion. These are associated to the particle-hole symmetry of the model, which in terms of the Dirac fermions (3.5) is given by [45, 47–49]

$$P = \prod_{i=1}^{N/2} \left(c_i + c_i^\dagger\right) K, \tag{3.13}$$

where $K$ is an anti-linear operator that maps $z \to \bar{z}$ for $z \in \mathbf{C}$.

For $N = 0$ (mod 8) this symmetry does not generate a protected degeneracy in the spectrum to begin with. If $N = 2, 6$ (mod 8), then $P$ maps the odd and even fermion parity sectors to each other, and there is no degeneracy on individual sectors, although there is one between sectors. Any of these cases follows the general considerations above for the generation of block-diagonal Haar random unitaries. For $N = 4$ (mod 8), however, there is a two-fold degeneracy on individual fixed-parity sectors. It is only in this case where the effective Hamiltonian has additional ground states with fixed fermion parity. To avoid this complication, we will implicitly restrict to $N \neq 4$ (mod 8) in what follows.

## 3.1 Time to design

As noted in [33], the first excited states of the effective Hamiltonian $H_k$ obtained in (3.4) for $p = 4$ are elementary fermion excitations above each ground state. This will also be true for general $p$. Consequently, we shall not attempt to use the large-$N$ collective mode description of this model in this paper, given that this method will not lead to the information of the spectral gap $\Delta_k$ and of the degeneracy of first excited states $N_*$ that we need. In what follows, we will examine the effective Hamiltonian (3.4) by direct diagonalization.

**Formation of a 1-design**

We shall begin with $k = 1$, in which case the effective Hamiltonian (3.4) can be written as

$$H_1 = J\left(1 - \psi_p^{1\bar{1}}\right), \qquad \psi_p^{1\bar{1}} \equiv \frac{1}{\binom{N}{p}} \sum_{i_1 < \ldots < i_p} \psi_{I_p}^1 \psi_{I_p}^{\bar{1}}. \tag{3.14}$$

Consider the set of states built from a single fermion excitation above any of the two ground states

$$|i; v\rangle = \psi_i^1 |W_e^v\rangle \qquad i = 1, ..., N, \quad v \in \{\pm\}. \tag{3.15}$$

Here $e \in \mathrm{Sym}(1)$ represents the trivial permutation, while $v$ represents the fermion parity of the TFD, $|W_e^v\rangle = |\infty, v\rangle_{1\bar{1}}$. Given that we can choose any $i \in \{1, ..., N\}$, there are a total of $N_* = 2N$ such states. Note that the value of $v$ in $|i; v\rangle_{1\bar{1}}$ only represents the fermion parity of the associated ground state, which is the opposite to the parity of $|i; v\rangle_{1\bar{1}}$ with respect to the 1 replica.

Commuting the single Majorana fermion with each term forming $\psi_p^{1\bar{1}}$ gives

$$\psi_{I_p}^1 \psi_i^1 = \psi_i^1 \psi_{I_p}^1 \times \begin{cases} -1 & \text{if } i \in I_p, \\ +1 & \text{otherwise}. \end{cases} \tag{3.16}$$

Therefore, commuting $\psi_p^{1\bar{1}}$ with $\psi_i^1$, there are $\binom{N-1}{p-1}$ terms in the sum over $I_p$ which contain $\psi_i^1$ and give a minus sign, and $\binom{N}{p} - \binom{N-1}{p-1}$ terms which do not contain $\psi_i^1$ and give a plus sign. After permuting the single fermion, we can use that $\psi_{I_p}^1 \psi_{I_p}^{\bar{1}} |W_e^v\rangle = |W_e^v\rangle$. The states (3.15) then correspond to the first excited states of the effective Hamiltonian (3.14), with energy given by the spectral gap $\Delta_1$,

$$H_1 |i, v\rangle = \Delta_1 |i, v\rangle, \qquad \Delta_1 = J\left(1 - \frac{\binom{N}{p} - 2\binom{N-1}{p-1}}{\binom{N}{p}}\right) = \frac{2Jp}{N}. \tag{3.17}$$

More generally, we can consider states of the form

$$\left| I_n^1, I_m^{\bar{1}}; v \right\rangle = \psi_{I_n}^1 \psi_{I_m}^{\bar{1}} |W_e^v\rangle, \tag{3.18}$$

where $\psi_{I_n}^1 \equiv \psi_{i_1}^1 ... \psi_{i_n}^1$ and $\psi_{I_m}^{\bar{1}} \equiv \psi_{i_1}^{\bar{1}} ... \psi_{i_m}^{\bar{1}}$ are Majorana strings. In order to keep track of the lineraly independent states, it is convenient to restrict to even values of $m$, so that the fermion parity of the $\bar{1}$ factor is given by $v$. There is a total of $N_{n,m} = \binom{N/2}{n}\binom{N/2}{m}$ linearly independent states of the form (3.18), where $n, m \leqslant N/2$. Given all possible choices of $n$ and even $m$, these states generate a complete basis of the double Hilbert space.

The states (3.18) are all eigenstates of the effective Hamiltonian,

$$H_1 \left| I_n^1, I_m^{\bar{1}}; v \right\rangle = E_{n,m} \left| I_n^1, I_m^{\bar{1}}; v \right\rangle, \qquad E_{n,m} = J\left(1 - \frac{f_n^p f_m^p}{\binom{N}{p}}\right), \tag{3.19}$$

where we have defined

$$f_n^p \equiv \sum_{\alpha=0}^{\min\{p,n\}} (-1)^\alpha \binom{n}{\alpha}\binom{N-n}{p-\alpha}. \tag{3.20}$$

The combinatorial factor $\binom{n}{\alpha}\binom{N-n}{p-\alpha}$ basically counts the number of Majorana strings in $\psi_p^{1\bar{1}}$ that share $\alpha$ Majorana fermions with a given $\psi_{I_n}^1$. The factor of $(-1)^\alpha$ arises from the commutation of these

Majorana strings with $\psi^1_{I_n}$. In Fig. 4 we plot the numerical spectrum of $H_1$, which is in complete agreement with our analytic formula (3.19).

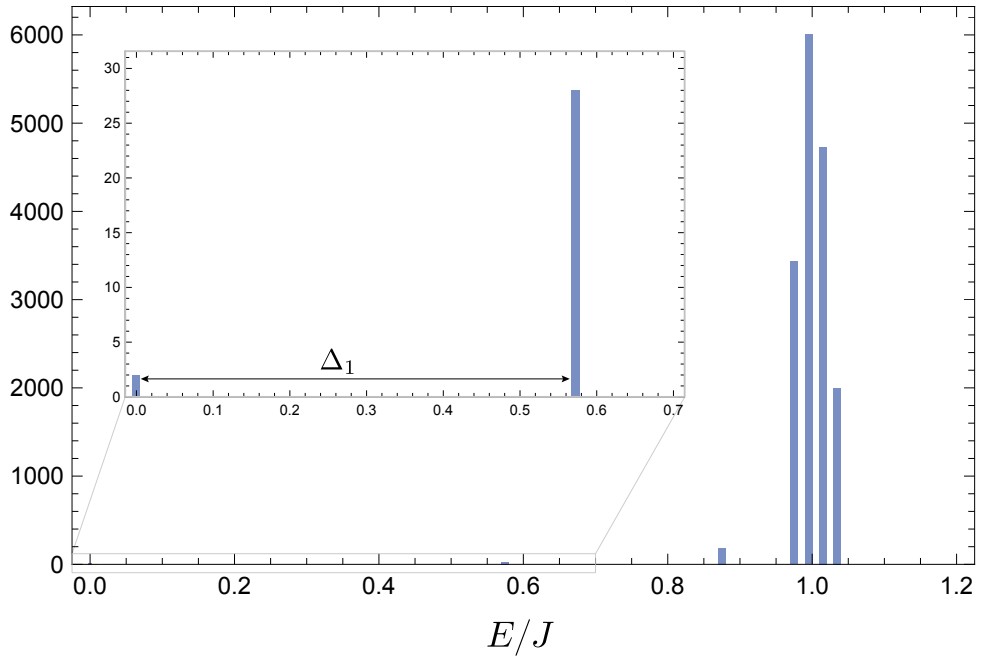

**Figure 4:** Spectrum of $H_1$ for $N = 14$ and $p = 4$. There are 2 ground states and $N_* = 28$ first excited states with energy $\Delta_1 = 8J/14$. Most of the states have energy $E_{n,m} \approx J$. At larger values of $N$, for $p \ll N$, the spectral gap $\Delta_1$ becomes parametrically smaller than $J$ and a tower of states with $E_{n,m} \approx \Delta_1(n + m)$ develops.

For $(n + m)p \ll N$, the low-lying excitations given by (3.19) are approximately additive

$$E_{n,m} \approx \Delta_1(n + m). \tag{3.21}$$

The partition function $Z_1(t) = \text{Tr}(e^{-tH_1})$ at late times is therefore approximately that of a collection of $N_{\text{eff}} = \gamma N/p$ free fermionic oscillators on top of each ground state, for some $\gamma \ll 1$,

$$\log Z_1(t) \approx \log 2 + N_{\text{eff}} \log\left(1 + e^{-\Delta_1 t}\right). \tag{3.22}$$

This gives a time to approximate 1-design of

$$t_1 \approx \Delta_1^{-1} \log \frac{1}{\left(1 + \frac{\varepsilon}{2}\right)^{\frac{1}{N_{\text{eff}}}} - 1} \approx \frac{N}{2Jp}\left(\log 2N + \log \gamma - \log p + \log \varepsilon^{-1}\right), \qquad p \ll N. \tag{3.23}$$

As a self-consistency check, it is easy to see that the rest of the states with energies close to $J$ provide a very small correction to the partition function at the time (3.23). We would have obtained a slightly worse approximation following the general considerations of section 2.3, and neglecting all but the first excited states.

Note that the approximation leading to (3.23) is only valid for $p \ll N$, for which the spectral gap $\Delta_1$ is parametrically smaller than $J$ and $N_{\text{eff}} \gg 1$, so that we only need to consider the tower of low-lying

excitations. For $p \sim N$, the spectral gap is precisely $\Delta_1 \sim J$, and so is the energy of approximately all the rest of the states. In this case, a better approximation is

$$Z_1(t) - Z_1(\infty) \approx (2^N - 2)e^{-Jt}, \tag{3.24}$$

which leads to

$$t_1 \approx \frac{1}{J} \left( N \log 2 + \log \varepsilon^{-1} \right), \qquad p \sim N. \tag{3.25}$$

We observe that, even though the gap gets larger and becomes $O(J)$ as $p$ is increased, the number of relevant states in the decay of the partition function also increases, which makes (3.25) parametrically large in $N$, even for the most non-local model. Of course, this scaling with system size is an artifact of choosing a $N$-independent normalization for the variance of the Hamiltonian in (3.3).

**Formation of a $k$-design**

Expanding the square, the effective Hamiltonian (3.4) can be recognized as

$$H_k = J \left( k - \sum_{r,s} \psi_p^{r\bar{s}} + (-1)^{\frac{p}{2}} \sum_{r<s} (\psi_p^{rs} + \psi_p^{\bar{r}\bar{s}}) \right), \tag{3.26}$$

where we have defined the fermion bilinears

$$\psi_p^{ab} \equiv \frac{1}{\binom{N}{p}} \sum_{i_1 < ... < i_p} \psi_{I_p}^a \psi_{I_p}^b, \qquad \text{for } a, b \in \{1, ..., k, \bar{1}, ..., \bar{k}\}. \tag{3.27}$$

Let us consider a given ground state $|W_\sigma^v\rangle$ of (3.26). The first excited states of $H_k$ are formed by single forward and backward replica excitations on top of this state

$$\left| I_n^r, I_m^{\sigma(\bar{r})}; \sigma, v \right\rangle = \psi_{I_n}^r \psi_{I_m}^{\sigma(\bar{r})} |W_\sigma^v\rangle. \tag{3.28}$$

Note that the fermion strings are placed on the forward and backward replicas of the same infinite-temperature TFD state in $|W_\sigma^v\rangle$.

Consider $\sigma = e$ for simplicity. From the Clifford algebra it follows that

$$\left[ \psi_p^{a\bar{b}} + \psi_p^{b\bar{a}} - (-1)^{\frac{p}{2}} \left( \psi_p^{ab} + \psi_p^{\bar{a}\bar{b}} \right) \right] |I_n^r, I_m^{\bar{r}}; e, v\rangle = 0, \tag{3.29}$$

for any $a, b \in \{1, ..., k\}$ with $a \neq b$. Moreover, we have that

$$\psi_p^{a\bar{a}} |I_n^r, I_m^{\bar{r}}; e, v\rangle = |I_n^r, I_m^{\bar{r}}; e, v\rangle, \qquad \forall a \in \{1, ..., k\}, \quad a \neq r, \tag{3.30}$$

$$\psi_p^{r\bar{r}} |I_n^r, I_m^{\bar{r}}; e, v\rangle = \frac{f_n^p f_m^p}{\binom{N}{p}} |I_n^r, I_m^{\bar{r}}; e, v\rangle, \tag{3.31}$$

were $f_n^p$ is defined in (3.20).

Then, it directly follows that such a state is an eigenstate of $H_k$ in (3.26), with eigenvalue

$$H_k \left| I_n^r, I_m^{\bar{r}}; e, v \right\rangle = E_{n,m} \left| I_n^r, I_m^{\bar{r}}; e, v \right\rangle, \qquad E_{n,m} = J \left( 1 - \frac{f_n^p f_m^p}{\binom{N}{p}} \right). \tag{3.32}$$

Given that $H_k$ is invariant under permutations of the backward replicas, this means that all the states (3.28) for general $\sigma \in \text{Sym}(k)$ are also eigenstates of $H_k$ with the same energy.

The gap is obtained from $E_{n,m}$ for $n = 1$ and $m = 0$

$$\Delta_k = \frac{2Jp}{N}, \tag{3.33}$$

which is independent of $k$, showing that this system satisfies the requirement 2 of section 2.3.

Again, for $(n + m)p \ll N$, the low-lying excitations given by (3.32) are approximately additive, and the partition function $Z_k(t) = \text{Tr}(e^{-tH_k})$ at late times is therefore approximately that of a collection of $N_{\text{eff}} = \gamma N/p$ free fermionic oscillators on top of each ground state, for some $\gamma \ll 1$. This gives

$$Z_k(t) \sim 2^k k! (1 + e^{-\Delta_k t})^{N_{\text{eff}}}, \tag{3.34}$$

which leads to a time to $k$-design

$$t_k \sim \frac{N}{2Jp} (k \log k - k + k \log 2 + \log \gamma N - \log p + \log \varepsilon^{-1}), \qquad p \ll N. \tag{3.35}$$

Up to logarithmic and subleading factors, we find that the time to design is linear in $k$, in complete agreement with the results of [33].

On the other hand, when $p \sim N$, all of the states (3.28) have approximately energy $J$. The number of states (3.28) per ground state is $k \times 2^{N-1}$, where the factor of $k$ comes from the different replicas in which a given ground state can be excited, and the factor of $2^{N-1}$ is the Hilbert subspace dimension generated by single-replica excitations (these excitations do not change the backward fermion parity of the TFD). A better approximation to the trace distance to design at sufficiently late times is in this case

$$Z_k(t) - Z_k(\infty) \sim 2^k k! k (2^{N-1} - 1) e^{-Jt}. \tag{3.36}$$

This leads to time to $k$-design

$$t_k \sim \frac{1}{J} \left( k \log k - k + (k + N - 1) \log 2 + \log k + \log \varepsilon^{-1} \right), \qquad p \sim N, \tag{3.37}$$

which, up to logarithmic and subleading factors, is also linear in $k$. The linear growth implies that the *time to approximate Haar random* $t_{\text{Haar}}$ (which is achieved for $k \gtrsim d$) in the completely non-local Brownian SYK model is exponential in the system size, $Jt_{\text{Haar}} \gtrsim O(2^{\frac{N}{2}})$.

## 3.2 Double-scaling limit

It is useful to take the double-scaling limit of the SYK model [45, 50–52] to see the transition between the $p$-local and the completely non-local regimes. The double-scaling limit is defined by

$$N \to \infty, \qquad \lambda \equiv \frac{2p^2}{N} \text{ fixed}, \qquad \mathsf{q} = e^{-\lambda}. \tag{3.38}$$

This simplifies the analysis substantially, given that in this regime the number of fermions in common between two Majorana strings $\psi_{I_p}$ and $\psi_{I'_{\tilde{p}}}$, denoted by $|I_p \cap I'_{\tilde{p}}|$, is Poisson distributed over the set of all $\psi_{I_p}$ of size $p$, with mean $\frac{p\tilde{p}}{N}$ (and similarly for $\psi_{I'_{\tilde{p}}}$). From the Clifford algebra,

$$\psi_{I_p} \psi_{I'_{\tilde{p}}} = (-1)^{|I_p \cap I'_{\tilde{p}}|} \psi_{I'_{\tilde{p}}} \psi_{I_p}. \tag{3.39}$$

Therefore, in the double scaling limit (3.38), when considering Majorana strings acting on $|W^v_\sigma\rangle$, we can replace

$$\psi_p^{r\bar{s}} \psi_{I_{\tilde{p}}}^r |W^v_\sigma\rangle = \mathsf{q}^{\Delta_{\tilde{p}}} \psi_{I_{\tilde{p}}}^r \psi_p^{r\bar{s}} |W^v_\sigma\rangle, \qquad \Delta_{\tilde{p}} = \frac{\tilde{p}}{p}. \tag{3.40}$$

where the permutation $\sigma$ satisfies $\sigma(r) = s$. The value of $\mathsf{q}^{\Delta_{\tilde{p}}}$ is essentially the mean value of the phase in (3.39) over the Poisson distribution.

### Formation of a 1-design

In the double scaling limit, we can use (3.40) to write down

$$\psi_p^{1\bar{1}} \psi_{I_n}^1 |W^v_e\rangle = \mathsf{q}^{\Delta_n} \psi_{I_n}^1 \psi_p^{1\bar{1}} |W^v_e\rangle, \qquad \Delta_n = \frac{n}{p}. \tag{3.41}$$

The operator $\psi_p^{1\bar{1}}$ can be interpreted as an 'average chord operator' connecting replicas 1 and $\bar{1}$ together, diagramatically represented by

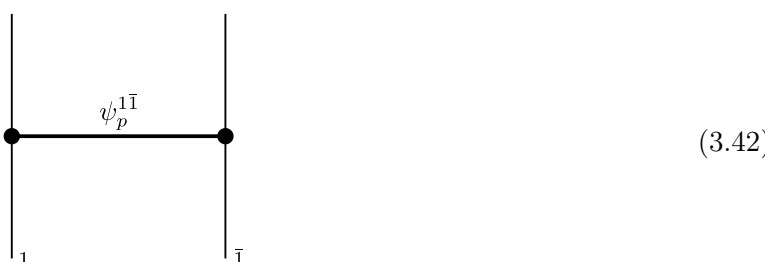

$$\tag{3.42}$$

Moreover, the identity (3.41) can be represented as the rule

$$
\tag{3.43}
$$

The infinite-temperature TFDs $|\infty, \pm\rangle_{1\bar{1}}$ are invariant under the action of the chord operator. This is represented diagramatically by the fact that the chord can contract when they act over these states

$$
\tag{3.44}
$$

For simplicity, we are omitting the details on the fermion parity of the TFDs, but a factor of $(-1)^{Q_r}$ needs to be included in order to talk about the two independent states. The chord operator is invariant under this symmetry.

From these diagrammatic rules, it is straightforward to see that the states $\left|I_n^1, I_m^{\bar{1}}; v\right\rangle$ defined in (3.18) are all eigenstates of the effective Hamiltonian $H_1$, with eigenvalue

$$
E_{n,m} = J(1 - \mathsf{q}^{\Delta_n + \Delta_m}) . \tag{3.45}
$$

Note that in the corresponding limit $n \to 1$ and $m \to 0$ this expression recovers the spectral gap $\Delta_1$ that we computed exactly in (3.33). In fact, (3.45) follows from the exact eigenvalue (3.20) by noting that

$$
\frac{\binom{n}{\alpha}\binom{N-n}{p-\alpha}}{\binom{N}{p}} \sim \frac{(\frac{pn}{N})^\alpha}{\alpha!} e^{-\frac{pn}{N}} , \tag{3.46}
$$

asymptotically approaches the Poisson distribution in the scaling limit in which $\frac{pn}{N}$ is kept finite.

Using (3.45) it is easy to see the different regimes. For finite $\lambda$, we have that

$$
E_{n,m} \approx
\begin{cases}
\Delta_1(n + m) & \text{for } n, m \ll p, \\
\\
J & \text{for } n + m \gg p .
\end{cases}
\tag{3.47}
$$

In this case there is a parametric separation between $\Delta_1$ and $J$ set by the double-scaling parameter $\lambda$. In particular, the number of allowed additive levels is controlled by

$$N_{\text{eff}} = \frac{J}{\Delta_1} - 1 = \frac{p - \lambda}{\lambda} \gg 1 \,. \tag{3.48}$$

The trace distance to 1-design is then determined by the $N_{\text{eff}}$ free fermion oscillators on top of each ground state and, additionally, the rest of the states with energy $J$,

$$Z_1(t) \approx 2(1 + e^{-\Delta_1 t})^{\frac{p-\lambda}{\lambda}} + (2^N - 2^{\frac{p}{\lambda}})e^{-Jt} \,. \tag{3.49}$$

This expression interpolates between (3.22) for finite $\lambda$ (with $\gamma \approx 1$) and (3.24) for the completely non-local limit $\lambda \to p$ (i.e. $p \to N/2$). Accordingly, the time to 1-design $t_1$ interpolates between (3.23) and (3.25) as a function of the degree of locality of the Hamiltonian.

**Formation of a $k$-design**

In a completely analogous way, the distance to $k$-design can be written in the double-scaling limit as

$$Z_k(t) \sim 2^k k! k \left(1 + e^{-\Delta_k t}\right)^{\frac{p-\lambda}{\lambda}} + 2^k k! k (2^{N-1} - 2^{\frac{p}{\lambda}})e^{-Jt} \,. \tag{3.50}$$

This expression interpolates between (3.34) for finite $\lambda$ (with $\gamma \approx 1$) and (3.36) in the non-local limit $\lambda \to p$. Accordingly, the time to $k$-design $t_k$ interpolates between (3.35) and (3.37) as a function of the degree of locality of the Hamiltonian.

## 3.3   Non-local limit

In the limit $\mathsf{q} \to 0$ ($\lambda \to \infty$) it is possible to solve the spectrum of the effective Hamiltonian exactly, given that all of the crossings are suppressed in this limit. In this case the chord operators become orthogonal projectors onto infinite-temperature TFD states

$$\lim_{\mathsf{q}\to 0} \psi_p^{ab} = \sum_{v=\pm} |\infty, v\rangle_{ab} \langle\infty, v|_{ab} \,, \tag{3.51}$$

for $a, b \in \{1, ..., k, \bar{1}, ..., \bar{k}\}$. For $a = r, b = \bar{s}$ this is represented by the diagram

$$\tag{3.52}$$

This identity follows from the fact that the infinite-temperature TFDs are invariant under the chord operator, while all other states, constructed by applying suitable Majorana strings on top of the infinite-temperature TFDs, will be annihilated after commuting the Majoranas with the chord operator.

In this limit, the effective Hamiltonian (3.26) becomes

$$
H_k = J \left( k - \sum_{r,s} |\infty, \pm\rangle_{r\bar{s}} \langle\infty, \pm|_{r\bar{s}} + (-1)^{\frac{p}{2}} \sum_{r<s} (|\infty, \pm\rangle_{rs} \langle\infty, \pm|_{rs} + |\infty, \pm\rangle_{\bar{r}\bar{s}} \langle\infty, \pm|_{\bar{r}\bar{s}}) \right) , \tag{3.53}
$$

where we are leaving the sum over $\pm$ implicit.

As we will see in the next section, this class of Hamiltonians arises naturally in completely non-local Brownian random matrix models. In particular, for time-reversal symmetric systems (even $\frac{p}{2}$) the effective Hamiltonian (3.53) exactly coincides with the large-$d$ approximation to the effective Hamiltonian (D.5) for the Brownian GOE model of random matrices that we develop in appendix D. The discrepancy between (3.53) and (D.5) comes from the approximation made in this section where we neglected subleading terms of the chord operator in the non-local limit.

As it will be clear later, Hamiltonians such as (3.53) can be exactly diagonalized using group theory. In particular, they are invariant under $\mathsf{SO}(d)$ transformations of the reference basis. The eigenspaces respect the decomposition of the Hilbert space into irreducible representations of $\mathsf{SO}(d)$ in the $2k$-fold tensor product of the fundamental representation. However, we will not attempt to do this here; following the analysis of the next section and Appendix D for (3.53) will yield a time to $k$-design which is precisely given by (3.37).

## 4  Brownian random matrix model

We will now introduce a random matrix model for the time-dependent Hamiltonian $H(t)$. The model is defined for a general Hilbert space $\mathcal{H}$ of dimension $d$. The intuitive idea, shown in Fig 1, is to independently select at each instant of time, a random Hamiltonian from some universality class of random matrices. This generates a completely unrestricted Brownian motion in $\mathsf{U}(d)$.[8] This ensemble is in some sense the most random dynamical evolution that one can select on a Hilbert space. Nevertheless, as we will show, the time to Haar is exponentially large in the entropy $Jt_{\text{Haar}} \gtrsim O(d)$ even in this case.

Formally, the ensemble of Hamiltonians is defined by the Euclidean matrix quantum mechanics

$$
\mathbb{E}\left[\bullet\right] = \frac{1}{\mathcal{N}} \int \mathcal{D}H(t)\, e^{-S}\left(\bullet\right), \qquad S = \frac{K}{2Jd} \int \mathrm{d}t\, \mathrm{Tr}\left(\frac{\dot{H}^2}{m^2} + H^2\right) . \tag{4.1}
$$

The normalization constant $\mathcal{N}$ is again determined by $\mathbb{E}\left[1\right] = 1$, while $K$ is a model-dependent constant that will be fixed below. For Brownian correlations, the matrix is taken to be infinitely massive $m \to \infty$

---

[8] The standard Brownian motion occurs in the Lie algebra $\mathfrak{u}(d)$. This induces a multiplicative Brownian motion on the group manifold $\mathsf{U}(d)$ by the rolling map [53].

so that the two-point function $\mathbb{E}\left[H_{ij}(t)H_{lm}(0)\right]$ is ultralocal in time.

To define the path integral measure $\mathcal{D}H(t)$ of the model, in this section we will consider the gaussian unitary ensemble (GUE) of random matrices. Choosing some arbitrary reference basis $\{|i\rangle\}_{i=1}^{d}$, the time-dependent Hamiltonian can be expressed as

$$H(t) = \sum_i x_{ii}(t)\Pi_{ii} + \sum_{i<j}\left(x_{ij}(t)\frac{\Pi_{ij}+\Pi_{ji}}{\sqrt{2}} + y_{ij}(t)\frac{\Pi_{ij}-\Pi_{ji}}{\sqrt{(-2)}}\right), \qquad \Pi_{ij} \equiv |i\rangle\langle j| . \tag{4.2}$$

for the real independent functions $x_{ij}(t)$ $(i \leqslant j)$ and $y_{ij}(t)$ $(i < j)$. With our normalization, all the Hermitian operators defining $H(t)$ in (4.2) have fixed variance in the maximally mixed state. We pick the natural path integral measure

$$\mathcal{D}H(t) \propto \prod_i \mathcal{D}x_{ii}(t)\prod_{i<j}\mathcal{D}x_{ij}(t)\mathcal{D}y_{ij}(t) . \tag{4.3}$$

That is, in the Brownian GUE model, the functions $\{x_{ij}(t), y_{ij}(t)\}$ are viwed as a collection of $K = d^2$ independent gaussian random white-noise correlated 'couplings' with zero mean and variance

$$\mathbb{E}\left[x_{ij}(t)x_{lm}(0)\right] = \mathbb{E}\left[y_{ij}(t)y_{lm}(0)\right] = \frac{Jd}{K}\delta(t)\delta_{il}\delta_{jm} . \tag{4.4}$$

The normalization in the variance of the couplings is chosen to match the general normalization used in section 2.1, where the variance of the Hamiltonian in the maximally mixed state $\rho_0 = \frac{1}{d}$ is taken to be

$$\mathbb{E}\left[\mathrm{Tr}(\rho_0 H(t)H(0))\right] = J\delta(t) . \tag{4.5}$$

From the general considerations of section 2.1, the $k$-th moment superoperator $\hat{\Phi}_{\mathcal{E}_t}^{(k)}$ of this ensemble can be written as the unnormalized thermal state (2.15) for the effective Hamiltonian

$$H_k = \frac{J}{4d}\sum_{i<j}\left[\left(\sum_{r=1}^{k}\Pi_{ij}^r + \Pi_{ji}^r - \Pi_{ij}^{\bar{r}} - \Pi_{ji}^{\bar{r}}\right)^2 - \left(\sum_{r=1}^{k}\Pi_{ij}^r - \Pi_{ji}^r + \Pi_{ij}^{\bar{r}} - \Pi_{ji}^{\bar{r}}\right)^2\right] +$$

$$+\frac{J}{2d}\sum_i\left(\sum_{r=1}^{k}\left[\Pi_{ii}^r - \Pi_{ii}^{\bar{r}}\right]\right)^2 . \tag{4.6}$$

Expanding this formula we recognize the effective Hamiltonian as

$$H_k = J\left(k - \sum_{r,s}|\infty\rangle_{r\bar{s}}\langle\infty|_{r\bar{s}}\right) + \frac{J}{d}\sum_{r<s}(\mathrm{SWAP}_{rs} + \mathrm{SWAP}_{\bar{r}\bar{s}}) . \tag{4.7}$$

for the operators

$$\text{SWAP}_{rs} = \sum_{i,j} \Pi^r_{ij} \Pi^s_{ji}, \qquad \text{SWAP}_{\bar{r}\bar{s}} = \sum_{i,j} \Pi^{\bar{r}}_{ij} \Pi^{\bar{s}}_{ji}, \tag{4.8}$$

$$|\infty\rangle_{r\bar{s}} \langle\infty|_{r\bar{s}} = \frac{1}{d} \sum_{i,j} \Pi^r_{ij} \Pi^{\bar{s}}_{ij}. \tag{4.9}$$

The operators $\text{SWAP}_{rs}$ and $\text{SWAP}_{\bar{r}\bar{s}}$ swap the corresponding replicas.

In the form (4.7), the effective Hamiltonian is clearly invariant under unitary transformations of the reference basis $|i\rangle \to V|i\rangle$ for $V \in \mathsf{U}(d)$. More specifically, the effective Hamiltonian satisfies

$$[H_k, V^{\otimes k} \otimes V^{*\,\otimes k}] = 0. \tag{4.10}$$

This symmetry is inherited from the choice of GUE measure (4.3). By the Schur-Weyl duality (see e.g. [18, 54, 55]), this symmetry severely restricts the $k$-th moment superoperator of the ensemble, and this is manifest in the form of the effective Hamiltonian (4.7), which only involves SWAP operators and orthogonal projectors into infinite-temperature TFDs.

This symmetry has further consequences. Most importantly for our purposes, the different eigenspaces of $H_k$ must all be left invariant by this symmetry, and fall into irreducible representations of $\mathsf{SU}(d)$ in the $k$-fold tensor product of the fundamental $\mathbf{d}$ and antifundamental $\overline{\mathbf{d}}$ representations, $\mathbf{d} \otimes \overline{\mathbf{d}} \otimes \ldots \otimes \mathbf{d} \otimes \overline{\mathbf{d}}$, which corresponds to how the symmetry acts on the replicated Hilbert space. It is not hard to see that such a representation contains $k!$ singlets for $k \leqslant d$. These states correspond to the states $|W_\sigma\rangle$ for $\sigma \in \text{Sym}(k)$ which linearly generate the ground space $\text{GS}_k$.

We remark, however, that the ensemble $\mathcal{E}_t$ is not right nor left invariant under unitary multiplication (as opposed to $\mathcal{E}_{\text{Haar}}$). By inspection of the effective Hamiltonian (4.7), this can be seen explicitly

$$H_k(V^{\otimes k} \otimes V^{*\,\otimes k}) \neq H_k. \tag{4.11}$$

In particular, the SWAP operators manifestly break this symmetry.

**Time to 1-design**

The effective Hamiltonian (4.7) for $k = 1$ is simply

$$H_1 = J\left(1 - |\infty\rangle_{1\bar{1}} \langle\infty|_{1\bar{1}}\right). \tag{4.12}$$

Naturally $|\infty\rangle_{1\bar{1}}$ is its only ground state, as expected from the general considerations of section 2.1. Moreover, the rest of the states have energy $\Delta_1 = J$. Therefore, the trace distance to a 1-design in this ensemble is

$$Z_1(t) - Z_1(\infty) = (d^2 - 1)\,e^{-Jt}. \tag{4.13}$$

The time to 1-design is then

$$t_1 = \frac{1}{J}\left(\log\!\left(d^2 - 1\right) + \log \varepsilon^{-1}\right).\tag{4.14}$$

Note that this matches the time to 1-design (3.25) for the Brownian $p$-SYK model in the non-local limit $p \sim N$ (for the value $d = 2^{\frac{N}{2}}$ in that case).

Moreover, it is convenient for later purposes to note that the eigenspaces of $H_1$ respect the decomposition

$$\mathbf{d} \otimes \overline{\mathbf{d}} = \mathbf{1} \oplus \left(\mathbf{d}^2 - \mathbf{1}\right),\tag{4.15}$$

of the product of the fundamental $\mathbf{d}$ and antifundamental $\overline{\mathbf{d}}$ of $\mathsf{SU}(d)$ into the adjoint $(\mathbf{d}^2 - \mathbf{1})$ (with eigenvalue $J$) and the singlet $\mathbf{1}$ (with zero eigenvalue). Explicitly, the Hilbert space of states

$$|\psi\rangle = \sum_{i,j} \psi_i^j\, |i,j\rangle_{1\bar{1}}\,,\tag{4.16}$$

decomposes into the traceless eigenspace $\sum_i \psi_i^i = 0$, in which case $\psi_i^j$ transforms in the adjoint of $\mathsf{SU}(d)$, and the pure trace eigenspace $\psi_i^j \propto \delta_i^j$, in which case the state $|\psi\rangle = |\infty\rangle_{1\bar{1}}$ is invariant.

**Time to 2-design**

Let us consider the case $k = 2$ for illustrative purposes, before providing a generalization for any $k$. The effective Hamiltonian (4.7) in this case reads

$$H_2 = J\left(2 - |\infty\rangle_{1\bar{1}}\langle\infty|_{1\bar{1}} - |\infty\rangle_{2\bar{2}}\langle\infty|_{2\bar{2}} - |\infty\rangle_{1\bar{2}}\langle\infty|_{1\bar{2}} - |\infty\rangle_{2\bar{1}}\langle\infty|_{2\bar{1}}\right) + \frac{J}{d}\left(\mathrm{SWAP}_{12} + \mathrm{SWAP}_{\bar{1}\bar{2}}\right).$$

$$\tag{4.17}$$

Consider a general state of the form

$$|\psi\rangle = \sum_{i,j} \psi_{i_1 i_2}^{j_1 j_2}\, |i_1, j_1, i_2, j_2\rangle_{1\bar{1}2\bar{2}}\,.\tag{4.18}$$

The eigenspaces of $H_2$ respect the decomposition of the Hilbert space

$$\mathbf{d} \otimes \overline{\mathbf{d}} \otimes \mathbf{d} \otimes \overline{\mathbf{d}} = \mathbf{1}^{\times 2} \oplus \left(\mathbf{d}^2 - \mathbf{1}\right)^{\times 4} \oplus \mathbf{A} \oplus \mathbf{M} \oplus \overline{\mathbf{M}} \oplus \mathbf{S}\,,\tag{4.19}$$

into two singlets $\mathbf{1}$, four adjoints $\mathbf{d}^2 - \mathbf{1}$, an antisymmetric traceless $\mathbf{A}$, two mixed $\mathbf{M}$ and a symmetric traceless $\mathbf{S}$ rank $(2,2)$ irreps of $\mathsf{SU}(d)$. These representations are constructed in a standard way using Young tableaux (see e.g. [56]). In Table 1 we classify them and compute the corresponding eigenvalue of $H_2$.

| irrep | $\mathbf{d^2 - 1}$ | $\mathbf{A}$ | $\mathbf{M}$ | $\mathbf{S}$ |
|---|---|---|---|---|
| $\psi^{j_1 j_2}_{i_1 i_2} =$ | $\delta^{i_1}_{j_1} \phi^{i_2}_{j_2}$ <br> $\left( \sum_i \phi^i_i = 0 \right)$ | $\psi^{j_1 j_2}_{[i_1 i_2]} = \psi^{[j_1 j_2]}_{i_1 i_2}$ <br> $\left( \sum_i \psi^{i j_2}_{i i_2} = 0 \right)$ | $\psi^{j_1 j_2}_{(i_1 i_2)} = \psi^{[j_1 j_2]}_{i_1 i_2}$ <br> $\left( \sum_i \psi^{i j_2}_{i i_2} = 0 \right)$ | $\psi^{j_1 j_2}_{(i_1 i_2)} = \psi^{(j_1 j_2)}_{i_1 i_2}$ <br> $\left( \sum_i \psi^{i j_2}_{i i_2} = 0 \right)$ |
| dimension | $d^2 - 1$ | $\dfrac{d^2(d+1)(d-3)}{4}$ | $\dfrac{(d^2-1)(d^2-4)}{4}$ | $\dfrac{d^2(d-1)(d+3)}{4}$ |
| $E$ | $\Delta_2 = J$ | $2J(1 - \frac{1}{d})$ | $2J$ | $2J(1 + \frac{1}{d})$ |

**Table 1:** The excited states of $H_2$ fall into irreps of $\mathsf{SU}(d)$ in the twofold tensor product of the fundamental and antifundamental. For simplicity, we only represent an example of adjoint and mixed irreps. The eigenvalues $E$ are approximately evenly spaced, where the gap is $\Delta_2 = J$ and the maximum eigenvalue is approximately $2\Delta_2$.

Consider the eigenspaces transforming in the adjoint, like for example

$$|\psi\rangle = |\infty\rangle_{1\bar{1}} \sum_{i,j} \psi^j_i \, |i,j\rangle_{2\bar{2}} \, , \tag{4.20}$$

where $\psi^j_i$ is traceless. These are eigenstates of (4.17) with eigenvalue

$$H_2 \, |\psi\rangle = \Delta_2 \, |\psi\rangle \, , \qquad \Delta_2 = J \, . \tag{4.21}$$

In total, the dimension of the subspace of states in the adjoint is $4(d^2 - 1)$, where the factor of 4 comes from all the possible adjoint irreps in the tensor product (4.19). One such representation is (4.20); the rest are constructed by suitable permutations of the replicas in the state (4.20).

Using the additional data collected in Table 1, the trace distance to 2-design in this ensemble corresponds to

$$Z_2(t) - Z_2(\infty) \approx 4(d^2 - 1) \, e^{-Jt} + (d^4 - 4d^2 + 2) e^{-2Jt} \, , \tag{4.22}$$

where we are neglecting energy separations of $O(J/d)$. This leads to a time to 2-design

$$t_2 \approx \frac{1}{J} \left( \log\big(d^2 - 1\big) + 2\log 2 + \log \varepsilon^{-1} \right) \, . \tag{4.23}$$

**Time to $k$-design**

For general $k \leqslant d$ we have the decomposition

$$\underbrace{\mathbf{d} \otimes \overline{\mathbf{d}} \otimes ... \otimes \mathbf{d} \otimes \overline{\mathbf{d}}}_{2k} = \mathbf{1}^{\times k!} \oplus (\mathbf{d}^2 - \mathbf{1})^{\times k! k} \oplus ... . \tag{4.24}$$

Denoting the $k$-fold product representation (4.24) by $R_k$, the trace distance to $k$-design can be written formally as

$$Z_k(t) - Z_k(\infty) = \sum_{\substack{\pi \in R_k \\ \pi \neq \mathbf{1}}} m_\pi \, d_\pi \, e^{-E_\pi t} . \tag{4.25}$$

where $\pi$ is an irreducible unitary representation of $\mathsf{SU}(d)$, $d_\pi$ is its dimension and $m_\pi$ is its multiplicity in $R_k$. The value of $E_\pi$ is the energy associated to the eigenspace transforming under $\pi$.[9]

The ground space $\mathrm{GS}_k$ is composed of the $k!$ singlet states $|W_\sigma\rangle$, which can be checked explicitly for (4.7). The first excited states transform in the adjoint and take the form

$$|\psi\rangle = |\infty\rangle_{1\bar{1}} ... |\infty\rangle_{k-1 \, \overline{k-1}} \sum_{i,j} \psi_i^j \, |i,j\rangle_{k\bar{k}} , \tag{4.26}$$

for $\psi_i^j$ traceless. The rest of the adjoint representations are constructed from (4.26) under suitable permutations of the replicas. It is easy to see explicitly that these states are eigenstates of $H_k$ with energy

$$H_k |\psi\rangle = \Delta_k |\psi\rangle , \qquad \Delta_k = J . \tag{4.27}$$

We thus find that the spectral gap is exactly independent of $k$ for this model. The number of first excited states is

$$N_* = k! k (d^2 - 1) , \tag{4.28}$$

where the factor of $k!$ comes from the number of ground states, and the factor of $k$ comes from the different replicas where each ground state can be excited to get (4.26). The number of first excited states $N_*$ scales exponentially with $k$ for large $k$. Therefore, this model satisfies the assumptions of section 2.3.

It then follows that the time to $k$-design behaves asymptotically as

$$t_k \sim \frac{1}{J} \left( k \log k - k + \log k + 2 \log d + \log \varepsilon^{-1} \right) . \tag{4.29}$$

Up to logarithmic or subleading factors, we find that $t_k$ grows linearly in $k$. This agrees with the timescale (3.37) for the non-local limit of the Brownian $p$-SYK model. Furthermore, this implies that

---

[9] By the Schur-Weyl duality the decomposition of $R_k$ into irreps of $\mathsf{SU}(d)$ also generates a natural decomposition of the Hilbert space into irreps of the permutation group $\mathrm{Sym}(k) \times \mathrm{Sym}(k)$ that swaps the forward and backward replicas separately. Since the effective Hamiltoninan (4.7) is invariant under such permutations, a fixed irrep $\pi$ must have the same eigenvalue, irrespectively of how it is represented on the Hilbert space.

time-to-Haar is exponential in the system size, $Jt_{\text{Haar}} \gtrsim O(d)$, even for the Brownian GUE model.

In fact, (4.29) is obtained in the crude approximation where we neglect all the higher excited states. This approximation will be reasonable when $k/d \ll 1$, since the energy of the rest of the states will be reasonably separated from the gap. However, when $k \approx d$, additional towers of states become close to the gap, and they make the time to $k$-design slighlty larger than (4.29). We can estimate the contribution as follows. By inspection of (4.7), the lowest energies among these states will be attained for the rank $(k, k)$ totally antisymmetric traceless irrep $\mathbf{A}_k$ of $\mathsf{SU}(d)$, which will have eigenvalue

$$E_{\mathbf{A}_k} = Jk \left( 1 - \frac{k-1}{d} \right) . \tag{4.30}$$

We see that for $k \leqslant d$ the energy of these states is always above the gap (4.27) (for $k > d$ this irrep is zero dimensional). However, when $k \approx d$ they are exponentially close to the gap. Similarly, it is not hard to construct states transforming in mixed irreps, with eigenvalue

$$E_{\mathbf{M}_k} = Jk \left( 1 - \frac{k-1-2n}{d} \right) , \tag{4.31}$$

For finite $n$ and $k \approx d$ the energy of these states will also be exponentially close to the gap.

To take into account these additional states, a better estimate of the time to design for $k \approx d$ is to consider an effective number of first excited states $N_*^{\text{eff}} = d^{2k\alpha}$, for some $O(1)$ constant $\alpha < 1$. Under this approximation, the time to $k$-design becomes

$$t_k \sim \frac{1}{J} \left( \alpha k \log d + \log \varepsilon^{-1} \right) , \tag{4.32}$$

which is paramterically larger than (4.29) and in fact exactly linear in $k$ in this regime.

In appendix D we extend our study in this section to Brownian gaussian random matrix models with instantaneous time-reversal and rotation symmetry (GOE) and symplectic ensembles (GSE).

## 4.1 Unrestricted Brownian motion in $\mathsf{U}(d)$

At first sight, the fact that even non-local time-dependent Hamiltonians take exponentially long to develop random unitaries might seem suprising. We will now provide an reformulation of this result from the point of view of classical Brownian motion.[10]

Individual time-dependent Hamiltonians $H(t)$ drawn from the Brownian GUE ensemble (4.1) determine a Brownian trajectory in the Lie algebra $\mathfrak{u}(d)$. From the rolling map [53], this motion induces a multiplicative Brownian motion in the unitary manifold $\mathsf{U}(d)$. This class of Brownian motion in compact Lie groups like $\mathsf{U}(d)$ has been extensively studied in probability and measure theory (see [53,57–61] and references therein). In fact, the results that we derived in this section are indirectly related to some of

---

[10] We thank Alexey Milekhin for valuable discussions on this point.

the integrals of [59].

The matrix stochastic differential equation defining this process in Stratonovich/Itô form is

$$dU(t) = -iU(t) \circ \mathrm{d}H(t) = -iU(t)\mathrm{d}H(t) + JU(t)\mathrm{d}t \,, \qquad U(0) = \mathbf{1} \,. \tag{4.33}$$

Here $J$ arises from our convention for the variance of the Brownian motion in $\mathfrak{u}(d)$. This Markov process can be decomposed into a standard Brownian motion of the global phase of $U(t)$ and an independent Brownian motion in $\mathsf{SU}(d)$. The latter is generated by $J\nabla_{\mathsf{SU}(d)}$, where $\nabla_{\mathsf{SU}(d)}$ is the Laplacian associated to the bi-invariant metric on $\mathsf{SU}(d)$. Recall that such metric is inherited from the Killing-Cartan form of the Lie algebra, which in our normalization reads $\langle iH_1, iH_2 \rangle = 2d\mathrm{Tr}(H_1H_2)$ for $iH_1, iH_2 \in \mathfrak{u}(d)$.

The formation of designs in this formulation of the Brownian GUE model is determined by the *heat kernel measure* $\rho_t(U)\mathrm{d}U$ on $\mathsf{U}(d)$, where $\mathrm{d}U$ is the Haar measure.[11] That is, the probability distribution defining the induced ensemble of unitaries $\mathcal{E}_t$ is the heat kernel

$$\frac{\partial \rho_t(U)}{\partial t} = J\nabla_{\mathsf{SU}(d)}\rho_t(U) \,, \tag{4.34}$$

initially concentrated at the identity, $\rho_0(U) = \delta(U - \mathbf{1})$. As time evolves, the distribution spreads out diffusively throughout $\mathsf{U}(d)$ until it becomes homogeneous. The endpoint is $\rho_\infty(U) = 1$, where the heat kernel measure becomes the Haar measure. The parameter $J$ plays the role of the diffusion constant.

The question is then how close $\rho_t(U)$ is from 1. By the Peter-Weyl theorem, the Hilbert space $L^2(\mathsf{U}(d))$ can be decomposed into the direct sum of (finite-dimensional) unitary irreps of $\mathsf{U}(d)$. Given that the bi-invariant Laplacian $\nabla_{\mathsf{SU}(d)}$ is the representation of the quadratic Casimir in $L^2(\mathsf{U}(d))$, these are invariant eigenspaces of the Laplacian. This allows to perform harmonic analysis and write the heat kernel as

$$\rho_t(U) = \sum_\pi \alpha_\pi \, f_\pi(U) \, \exp(-c_\pi Jt) \,, \tag{4.35}$$

where $f_\pi(U)$ is an eigenfunction of the Laplacian with eigenvalue $c_\pi$, corresponding to the value of the Casimir in the irreducible representation $\pi$. The eigenfunctions satisfy the orthonormality conditions,

$$\int \mathrm{d}U \, f_\pi(U) \, f_{\pi'}^*(U) = \delta_{\pi,\pi'} \,. \tag{4.36}$$

Therefore, the coefficients $\alpha_\pi$ in the expansion (4.35) are just determined by the initial condition in the form

$$\alpha_\pi = f_\pi^*(\mathbf{1}) = 1 \,, \tag{4.37}$$

where the fact that these coefficients can always be set to 1 is a matter of the definition of $f_\pi(U)$.

For the trivial representation $\pi = \mathbf{1}$ the eigenfunction is constant, $f_{\mathbf{1}}(U) = 1$, and the eigenvalue vanishes, $c_{\mathbf{1}} = 0$. The rest of the modes for $\pi \neq \mathbf{1}$ decay in time. Given our normalization of the

---

[11] For simplicity, we have omitted the probability distribution of the global phase.

metric, the value of the Casimir is $\min_{\pi \neq \mathbf{1}}\{c_\pi\} = O(d^0)$. For instance, the values of the Casimir in the fundamental (or antifundamental) and the adjoint irreps are

$$c_{\mathbf{d}} = c_{\bar{\mathbf{d}}} = \frac{d^2 - 1}{2d^2}, \qquad c_{\mathbf{d^2}-\mathbf{1}} = \frac{1}{2}. \tag{4.38}$$

From these considerations, the $L^2$ norm distance between the heat kernel measure and the Haar measure can be computed as

$$\left\| \rho_t - 1 \right\|_2^2 = \sum_{\pi \neq \mathbf{1}} e^{-2c_\pi J t}. \tag{4.39}$$

We shall now provide a rough estimate the decay of (4.39). Given that the difference in quadratic Casmilir $c_\pi$ is $O(d^{-2})$ for neighbouring irreps, there will be $O(d^2)$ irreps with eigenvalue $2c_\pi \approx 1$, and we will neglect the contribution of the rest. Therefore, the time it for the heat kernel measure to become approximately homogeneous in $L^2$ norm, $t_{L^2} = \inf\{t : \left\| \rho_t - 1 \right\|_2 < \varepsilon\}$, is

$$t_{L^2} \sim \frac{2}{J} \left( \log d + \log \varepsilon^{-1} \right). \tag{4.40}$$

This time is polynomial in the entropy, $\log d$, which is what we could have expected from the intuitive picture of diffusion on a compact space. The fact that (4.40) scales with the entropy is only a consequence of our choice of non-extensive normalization for the Brownian step.

The reason why this timescale is parametrically smaller than the time to Haar computed in this paper is that the definition of an approximate $k$-design (2.5) is a much stronger requirement than $\left\| \rho_t - 1 \right\|_2 < \varepsilon$. Mainly, two probability distributions in $\mathsf{U}(d)$ can be $\varepsilon$ close in $L^2$ norm distance, but they might have $k$-th moment superoperators which are not $\varepsilon$ close in trace distance, for large enough values of $k$.

To see this, note that the $k$-th moment superoperator in this picture is simply[12]

$$\hat{\Phi}_{\mathcal{E}_t}^{(k)} = \int \mathrm{d}U \, \rho_t(U) \, U^{\otimes k} \otimes U^{* \, \otimes k} = e^{-t H_k}. \tag{4.41}$$

Using the harmonic decomposition of the heat kernel (4.35), this superoperator can also be written as

$$\hat{\Phi}_{\mathcal{E}_t}^{(k)} = \sum_\pi e^{-c_\pi t J} \, \mathcal{O}_\pi^{(k)}, \qquad \mathcal{O}_\pi^{(k)} \equiv \int \mathrm{d}U \, f_\pi(U) \, U^{\otimes k} \otimes U^{* \, \otimes k}. \tag{4.42}$$

Note that for $\pi = \mathbf{1}$, the superoperator is $\mathcal{O}_{\mathbf{1}}^{(k)} = \hat{\Phi}_{\mathcal{E}_{\mathrm{Haar}}}^{(k)}$. For $k \neq 1$ the rest of the superoperators commute with $\mathcal{O}_{\mathbf{1}}^{(k)}$, but generally they need not commute with each other. For this reason, this formulation is less useful than the one in terms of the effective Hamiltonian $H_k$ when it comes to the computation of the decay of the trace distance to design.

Let us however use the fact that from section 2.2 we know the trace distance to design is given in

---

[12] We can neglect the probability distribution of the global phase of $U$, since it does not affect the moment superoperator.

terms of the partition function (2.23) to write it as

$$\left\| \hat{\Phi}^{(k)}_{\mathcal{E}_t} - \hat{\Phi}^{(k)}_{\mathcal{E}_{\text{Haar}}} \right\|_1 = \sum_{\pi \neq \mathbf{1}} e^{-c_\pi t J} \operatorname{Tr}\left( \mathcal{O}^{(k)}_\pi \right) . \tag{4.43}$$

The value of $\operatorname{Tr}\left( \mathcal{O}^{(k)}_\pi \right)$ can also be related to a higher moment of the SFF in $\mathsf{U}(d)$, in the vein of the discussion of section 2.4.

Given that $\operatorname{Tr}(\mathcal{O}^{(k)}_{\mathbf{1}}) = k!$, it is reasonable to expect that the trace in (4.43) scales at least as $k!$,[13] so that the time that it takes for an approximate $k$-design to form is

$$t_k \sim \frac{1}{J}(k \log k - k + \varepsilon^{-1}) \gg t_{L^2} . \tag{4.44}$$

That is, the additional factor of the trace in (4.43), associated to the $k$-th moments of the eigenfunctions $f_\pi(U)$, makes the time to $k$-design parametrically larger in $k$ than the time for the heat kernel to become approximately homogeneous in $L^2$ norm.

## 5 Complexity growth vs the generation of randomness

In the previous sections we have shown that the formation of approximate $k$-designs takes a time which is generally $Jt_k = O(k)$, even for ensembles of highly non-local time-dependent Hamiltonians. A natural question based on the study of [18, 38, 45] is whether this has any implications for the complexity of the time-evolution operator $U(t)$ in the circuit model of quantum computation.

We shall consider the Hilbert space $\mathcal{H}$ of $N$ qubits, of dimension $d = 2^N$, and a universal finite few-body gate set $\mathsf{G}$. Let $\mathcal{E}_k$ be an approximate $k$-design and let $U \in \mathcal{E}_k$. From a counting argument, it follows that the circuit complexity $\mathcal{C}(U)$ is lower bounded by [18]

$$\mathcal{C}(U) \gtrsim \frac{2kN - k \log k + k}{\log(|\mathsf{G}|N^2)} . \tag{5.1}$$

We can now define the completely non-local Brownian GUE model in this Hilbert space. Using the results of section 4 we conclude that the circuit complexity of time-evolution in this system satisfies

$$\mathcal{C}(U(t)) \gtrsim \frac{Jt\,(2N - \log Jt + 1)}{\log(|\mathsf{G}|N^2)} . \tag{5.2}$$

The right hand side seems to provide a linear growth in circuit complexity for exponentially long times, up to logarithmic factors.

By now it should be obvious that (5.2) cannot be a tight bound, at the very least for non-local

---

[13] A way to justify this is that the average value gives $\sum_\pi \operatorname{Tr}(\mathcal{O}^{(k)}_\pi) = d^{2k}$ which scales exponentially in $k$. We expect that the representations which are close to $\mathbf{1}$ in value of the Casimir $c_\pi$ to have similar features for the moments; that is why we assumed a more modest $k!$ scaling instead.

systems.[14] Otherwise, it would imply that exponentially powerful computations are rarely generated by the set of non-local time-dependent Hamiltonians. From general considerations, the left hand side of (5.2) should saturate at early times $Jt = \text{poly}(N)$, while on the other hand, the the right hand side simply keeps growing until $Jt = \text{poly}(2^N)$ times.

More concretely, let us sketch where the bound (5.1) arises from. Let $\mathcal{U}(\mathcal{C})$ be the set of unitaries which up to some resolution in trace distance have complexity $\mathcal{C}$. The bound (5.1) was derived in [18] from the condition that for any $k$-design $\mathcal{E}_k$ the number of elements $|\mathcal{E}_k|$ up to some resolution in trace distance must be lower bounded. Naturaly, if the set $\mathcal{U}(\mathcal{C})$ contains fewer elements than a $k$-design, it will not be able to contain the $k$-design

$$|\mathcal{U}(\mathcal{C})| < |\mathcal{E}_k| \quad \Rightarrow \quad \mathcal{E}_k \nsubseteq \mathcal{U}(\mathcal{C}) \,, \tag{5.3}$$

and therefore there must exist unitaries $U \in \mathcal{E}_k$ with $\mathcal{C}(U) > \mathcal{C}$.

Let us consider the smallest value of the complexity for which the neccessary condition $|\mathcal{U}(\mathcal{C})| \geqslant |\mathcal{E}_k|$ is satisfied

$$\mathcal{C}_{\min}(k) \equiv \min\{\mathcal{C} : |\mathcal{U}(\mathcal{C})| \geqslant |\mathcal{E}_k|\} \,. \tag{5.4}$$

Then it must be true that the maximum complexity that a unitary $U \in \mathcal{E}_k$ can have is at least $\mathcal{C}_{\min}$

$$\max_{U \in \mathcal{E}_k} \mathcal{C}(U) \geqslant \mathcal{C}_{\min}(k) \,. \tag{5.5}$$

The value of $\mathcal{C}_{\min}(k)$ is basically the right hand side of (5.1), where the minimum value of $|\mathcal{E}_k|$ has been estimated from the frame potential.[15]

However, our results for the Brownian random matrix model suggest that $\mathcal{C}(U) \gg \mathcal{C}_{\min}$ for most of $U \in \mathcal{E}_k$, with an exponential difference in the scaling with $N$, even for small values of $k$. That is, we expect that most unitaries $U \in \mathcal{E}_k$ for an approximate $k$-design $\mathcal{E}_k$ like the one studied in section 4 have maximal circuit complexity.[16] These unitaries must provide a very atypical and small sample of all the unitaries with maximal complexity, since they are not drawn from the Haar distribution.

## 5.1 Brownian motion and complexity growth

The linear growth of complexity for random circuits is expected from the fact that the occurrence of shortcuts at polynomial circuit time is extremely rare. We shall now provide a geometric argument for why a smooth version of the linear growth in complexity is expected for continuous $p$-local Brownian systems. Moreover, we will also motivate the hyperfast growth of complexity for non-local Brownian systems from this point of view.

---

[14] This is also expected to be true for time-independent non-local Hamiltonians, like the systems studied in [32]; see e.g. [62] pointing in this direction.

[15] The left hand side of (5.1) should include a maximum over $U \in \mathcal{E}_k$.

[16] Similar considerations should hold for strong complexity, following Theorem 9 in [38].

In order to do this, we will adapt the model of complexity geometry of [63–65] and study Brownian motion on this metric. In complexity geometry, the linear growth follows from a nice property of Brownian motion on negatively curved spaces: the radial spread is ballistic, and approximately optimal compared to the geodesic (see Fig. 5).[17]

*Restricted Brownian motion*

In order to make the argument, we shall consider $\mathsf{SU}(d)$ endowed with an oversimplified toy model of the complexity metric $g_{\mu\nu}$. To illustrate our main point, we shall consider the analysis locally close to $1$ and restrict to the simplest negatively curved metric

$$\mathrm{d}s^2 = \mathrm{d}\mathcal{C}^2 + \sinh^2(\mathcal{C})\,\mathrm{d}\Omega_{d_p-1}^2 + d^2\,\mathrm{d}s_{d-d_p}^2\,, \tag{5.6}$$

Here $\mathcal{C}$ is simply the complexity in the $p$-local directions, measured as the geodesic proper distance to the identity operator, at $\mathcal{C} = 0$. The second factor $\mathrm{d}s_{d-d_p}^2$ represents the exponentially penalized $d - d_p$ directions of the tangent space, associated to non-local Hamiltonians, where $d_p$ is the number of $p$-local directions. This model of complexity geometry captures that the number of unitaries in the $p$-local directions with complexity $\mathcal{C}$ (in some tessellation of the Poincaré ball) grows exponentially with $\mathcal{C}$.

We will model the $p$-local Brownian ensemble $\mathcal{E}_t$ by the probability measure associated to Brownian motion on the metric (5.6). This is the heat kernel measure $\rho_t\mathrm{dVol}(g)$ in the metric (5.6), where $\mathrm{dVol}(g)$ is the volume element of the complexity metric $g_{\mu\nu}$, and

$$\frac{\partial \rho_t}{\partial t} = D\nabla^2 \rho_t\,, \qquad \rho_0(U) = \delta(U-1)\,. \tag{5.7}$$

Here $D = J/d_p$ plays the role of the diffusion constant, and $\nabla^2$ is the Laplacian of (5.6). The factor of $d_p$ is necessary to match our normalization in the previous sections (i.e. note that the eigenvalues of the Laplacian in section 4.1 are $O(1)$ numbers in our convention of the bi-invariant metric).

Under the assumption of spherical symmtery $\rho_t = \rho_t(\mathcal{C})$ in the $p$-local directions, and neglecting the extremely slow motion in the non-local directions, the diffusion equation (5.7) reduces to

$$\frac{\partial \rho_t}{\partial t} = D\left(\partial_{\mathcal{C}}^2 + \frac{d_p-1}{r}r'\partial_{\mathcal{C}}\right)\rho_t\,. \tag{5.8}$$

Exact analytic solutions are known, see e.g. [68]. For our purposes, however, it is enough to consider the approximation $\mathcal{C} \gg 1$, where $r'/r \approx 1$ and the equation (5.8) reduces to a one-dimensional diffusion equation under the change of spatial variable $\mathcal{C} \to \mathcal{C} + Jt$. The approximate solution is then

$$\rho_t(\mathcal{C}) \approx \frac{\alpha}{(4\pi Dt)^{1/2}} \exp\left(-\frac{(\mathcal{C}+Jt)^2}{4tD}\right)\,, \tag{5.9}$$

---

[17] This mathematical fact is responsible of fast scrambling of local systems on expander graphs [66, 67].

where $\alpha^{-1} \approx \text{Vol}(\mathbf{S}^{d-1})$ is a normalization constant.

The radial probability distribution $p_t(\mathcal{C}) \equiv \alpha^{-1}\rho_t(\mathcal{C})\sinh(\mathcal{C})^{d_p-1}$ gives the expected value

$$\mathbb{E}[\mathcal{C}] \equiv \int_0^\infty \mathrm{d}\mathcal{C}\, p_t(\mathcal{C}) \approx Jt\,, \tag{5.10}$$

We see that the radial proper distance, or complexity, grows ballistically, at a rate given by $J$, until $Jt_{\text{sat}} = O(d^2)$, where a cutoff surface is placed in this model. The non-extensivity of the slope with the entropy is a consequence of the non-extensive normalization for the Brownian step chosen throughout this paper.

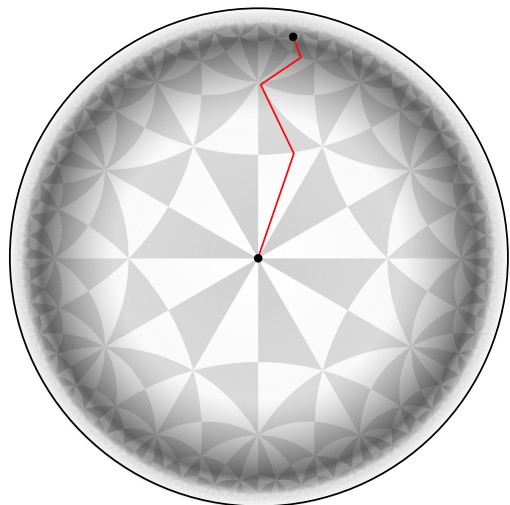

**Figure 5:** Brownian motion on hyperbolic space. The radial spread is ballistic, and approximately optimal compared to the geodesic.

Therefore, our argument suggests that circuit complexity grows linearly for very long times for $p$-local Brownian systems, and the linear bound provided by (5.1) is parametrically relevant, as illustrated on the left plot of Fig. 2. The slopes might differ in $O(1)$ constants; they will generally depend on the precise definition of quantum complexity, such as the particular choice of universal gate set.

*Unrestricted Brownian motion*

As explained in section 4.1, for the non-local Brownian GUE model, the measure $\rho_t(U)\mathrm{d}U$ defining the unitary ensemble $\mathcal{E}_t$ corresponds to the heat kernel measure in the bi-invariant metric of $\mathsf{SU}(d)$. From the form of (4.39) we see that heat kernel relaxes at early times to the Haar distribution in $L^2$ norm, in a timescale set by (4.40). Given this, the average value of the proper distance to the identity in complexity geometry will saturate to its Haar (maximum) value at early times,

$$\mathbb{E}[\mathcal{C}] \approx \mathcal{C}_{\text{Haar}} = O(d^2)\,, \qquad \text{for } t \gtrsim t_{\text{sat}} = J^{-1}\log d^2\,. \tag{5.11}$$

Therefore, we have motivated the hyperfast growth of complexity illustrated in Fig. 2, in contrast to the slow generation of randomness, which takes a time $Jt_{\text{Haar}} \gtrsim O(d)$ for these systems.

## 6    Conclusions

In this paper we have studied the dynamical generation of randomness in chaotic many-body quantum systems, as a function of the degree of locality of the time-dependent Hamiltonian. We have shown that for Brownian systems, which resemble continuous versions of random circuits, the growth in design is universally linear both for $p$-local and non-local systems. Based on this observation, we have conjectured that, generally, unless a large degree of fine-tuning is present in the ensemble of time-dependent Hamiltonians, it is not possible to dynamically generate random unitaries efficiently. We have pointed out the distinction between complexity and randomness by providing examples of rather general non-local systems which are exponentially complex to implement on a conventional quantum computer, but which are still slow generators of randomness.

For Brownian random matrix Hamiltonians defined by the GUE path integral measure, we have pointed out that the probability distribution of the ensemble is defined by the heat kernel measure in $\mathsf{SU}(d)$. The relaxation timescale of the probability distribution in $L^2$ norm is itself $O(\log d)$; however, the timescale for the moments to relax is enhanced by the large value of the moments. Together with the linear growth in design for $p$-local systems, our results suggests that the global relaxation of the probability distribution defining the ensemble of unitaries is insufficient for the generation of randomness.

An interesting open question is whether our conjecture can itself be formulated at the level of the Hilbert space. Definitely if $U$ is Haar random, $U |\psi_1\rangle$ is a random state; however, the converse is not true. For this reason it might be easier to dynamically generate distributions of random states, relative to some reference state $|\psi_1\rangle$, with highly non-local Hamiltonians. For instance, Hamiltonians of similar form to (1.1) for $|\psi_2\rangle$ chosen randomly, all map $|\psi_1\rangle$ to a random state. However the unitaries that these Hamiltonians generate are themselves far from Haar random, since each of them contains a two-dimensional invariant subspace.

In this context, it would also be interesting to determine how long it takes for a many-body quantum system to dynamically develop the moments of the Porter-Thomas distribution for the return probabilities of states in a bit string basis. Finally, given that partial projective measurements allow for the efficient generation of random states [69–71], another interesting question is whether there is a way to approximately generate Haar random unitaries via measurement of collections of states efficiently.

In a different direction, it would be interesting to elucidate the holographic manifestation of the dynamical generation of $k$-randomness studied in this paper for systems describing black holes, in situations where these systems are perturbed with suitable time-dependent perturbations. Since the generation of randomness studied in this paper is universal and it persists until very late times after scrambling, it is reasonable to expect that randomness should be reflected in properties of the black hole interior, in the vein of holographic complexity [72, 73].

Furthermore, non-local systems, and in particular the double-scaled SYK model, has been recently proposed as a microscopic realization of de Sitter space holography [74–77]. In this correspondence, the hyperfast growth of the computational complexity of the time-evolution operator is conjectured to be captured by the behavior of maximal volume slices anchored to the stretched cosmological horizons for antipodal observers [74]. Based on the results of this paper, it would be interesting to investigate whether there is a holographic inprint of the slow generation of randomness, after perturbing the cosmological horizon of de Sitter space with suitable noise. We leave these problems for future investigation.

## Acknowledgments

We thank Stefano Antonini, José Barbón, Greg Bentsen, Luca Iliesiu, Shao-Kai Jian, Javier Magán, Alexey Milekhin, John Preskill, Sreeman Reddy Kasi Reddy and Tiangang Zhou for conversations. MS is grateful to the Walter Burke Institute for Theoretical Physics at the California Institute of Technology for their hospitality during the final stages of this work. We acknowledge support from the U.S. Department of Energy through DE-SC0009986 and QuantISED DE-SC0020360. This preprint is assigned the code BRX-TH-6720.

## A    Moment maps of the Haar ensemble

In this appendix, we recall some basic facts about the $k$-th moment superoperators for the Haar ensemble $\mathcal{E}_{\mathrm{Haar}}$ and block-diagonal Haar ensemble $\mathcal{E}_{\mathrm{Haar}_G}$ in the case that there exists a global symmetry $G$.

The superoperator representation of the quantum channel $\Phi_{\mathrm{Haar}}^{(k)}$ is explicitly given by

$$\hat{\Phi}_{\mathrm{Haar}}^{(k)} \equiv \int_{\mathsf{U}(d)} \mathrm{d}U \, U^{\otimes k} \otimes U^{*\,\otimes k} \,, \tag{A.1}$$

where $\mathrm{d}U$ is the Haar measure in $\mathsf{U}(d)$. From the right invariance of the Haar measure, it follows that $\hat{\Phi}_{\mathrm{Haar}}^{(k)}$ is invariant under right multiplication

$$\hat{\Phi}_{\mathrm{Haar}}^{(k)} \left( V^{\otimes k} \otimes V^{*\,\otimes k} \right) = \hat{\Phi}_{\mathrm{Haar}}^{(k)}, \qquad \forall \, V \in \mathsf{U}(d) \,. \tag{A.2}$$

Analogously, $\hat{\Phi}_{\mathrm{Haar}}^{(k)}$ is invariant under left multiplication.

This symmetry severely restricts the form of the superoperator $\hat{\Phi}_{\mathrm{Haar}}^{(k)}$. In particular, $\hat{\Phi}_{\mathrm{Haar}}^{(k)}$ must only act non-trivially on the singlet states of $k$-th tensor product of the fundamental and antifundamental irreducible representations of $\mathsf{SU}(d)$ (i.e. in the representation $V^{\otimes k} \otimes V^{*\,\otimes k}$ associated to how the symmetry acts on the total Hilbert space). By the Schur-Weyl duality, the invariant states generate a natural representation of the symmetry group $\mathrm{Sym}(k)$. They are explicitly characterized by

$$|W_\sigma\rangle = \bigotimes_{r=1}^{k} |\infty, v\rangle_{r\sigma(\bar{r})} \,, \qquad \sigma \in \mathrm{Sym}(k) \,. \tag{A.3}$$

The permutation $\sigma \in \text{Sym}(k)$ labels the way to assign infinite-temperature TFDs between forward and backward replicas (here labeled by $r$ and $\bar{s} = \sigma(\bar{r})$). For $k \leqslant d$, these states generate a $k!$-dimensional subspace $V_k$ of the total Hilbert space.

Given that $(\hat{\Phi}^{(k)}_{\text{Haar}})^2 = \hat{\Phi}^{(k)}_{\text{Haar}}$, and that $\hat{\Phi}^{(k)}_{\text{Haar}} |W_\sigma\rangle = |W_\sigma\rangle$, the Haar superoperator is the orthogonal projector to the space generated by the invariant states,

$$\hat{\Phi}^{(k)}_{\text{Haar}} = \Pi_{V_k} . \tag{A.4}$$

This is essentially what we used in (2.20).

**Moment maps for the block-diagonal Haar ensemble**

In case that there exists a global symmetry $G$, the Hilbert space decomposes into $n_G$ superselection sectors $\mathcal{H} = \oplus_g \mathcal{H}_g$, where $G = \oplus_g (g \times 1_g)$ takes the value $g$ on each sector $\mathcal{H}_g$. We can define the block-diagonal Haar measure $\prod_g \mathrm{d}U_g$ for unitaries which respect this decomposition, $[U, G] = 0$. Such unitaries can be decomposed into smaller unitaries $U = \oplus_g U_g$, where $U_g \in U(d_g)$ for $d_g = \dim(\mathcal{H}_g)$. We will assume that $d_g > 1 \ \forall g$.

The block-diagonal Haar superoperator is

$$\hat{\Phi}^{(k)}_{\text{Haar}_G} \equiv \int_{[U,G]=0} \mathrm{d}U_{g_1}...\mathrm{d}U_{g_{n_G}} \, U^{\otimes k} \otimes U^{* \, \otimes k} . \tag{A.5}$$

The symmetry of $\hat{\Phi}^{(k)}_{\text{Haar}_G}$ is now reduced to right (or left) multiplication of the form (A.2) by any block diagonal $V \in \mathsf{U}(d)$, restricted to $[V, G] = 0$. This symmetry again restricts the form of the superoperator to act only on the invariant states of this symmetry. Now, since the symmetry group is smaller, there will be more invariant states.

The total Hilbert space of $2k$ replicas where $\hat{\Phi}^{(k)}_{\text{Haar}_G}$ acts can be decomposed into the direct sum of subspaces of the form $\mathcal{H}_{g_1} \otimes ... \otimes \mathcal{H}_{g_{2k}}$, where symmetry $V$ acts on the last $k$-factors by complex conjugation. These Hilbert subspaces are invariant under the symmetry. In such subspaces, if the number of fundamental and antifundamental representations of some symmetry subgroup $V_g$ (acting non-trivially only on $\mathcal{H}_g$) is different, there cannot be singlets. Therefore, singlet states exist in $\mathcal{H}_{g_1} \otimes ... \otimes \mathcal{H}_{g_{2k}}$ provided that the first $k$ global charges $\{g_1, ..., g_k\}$ can be identified with the last $k$ global charges $\{g_{k+1}, ..., g_{2k}\}$ in groups of even number of elements. For each such group of $2l$ factors with the same charge, by the Schur-Weyl duality, the singlets generate a natural representation of the permutation group $\text{Sym}(l)$ group on these $2l$ factors.

That is, to label the invariant states, we can choose a permutation $\sigma \in \text{Sym}(k)$ that identifies the forward and backward replicas in pairs of infinite-temperature TFDs, and then assign some global charge

$v = \{g_1, ..., g_k\}$ to each TFD. This fully determines all the invariant states

$$|W_\sigma^v\rangle = \bigotimes_{r=1}^{k} |\infty, v\rangle_{r\sigma(\bar{r})} , \qquad \sigma \in \mathrm{Sym}(k) , \qquad v = \{g_1, ..., g_k\} . \tag{A.6}$$

The invariant states generate a $n_g^k k!$-dimensional subspace

$$V_k^G = \mathrm{Span}\{|W_\sigma^v\rangle : \sigma \in \mathrm{Sym}(k), v = \{g_1, ..., g_k\}\} . \tag{A.7}$$

Given that $(\hat{\Phi}_{\mathrm{Haar}_G}^{(k)})^2 = \hat{\Phi}_{\mathrm{Haar}_G}^{(k)}$, and that $\hat{\Phi}_{\mathrm{Haar}_G}^{(k)} |W_\sigma^v\rangle = |W_\sigma^v\rangle$, the Haar superoperator is the orthogonal projector to this subspace,

$$\hat{\Phi}_{\mathrm{Haar}_G}^{(k)} = \Pi_{V_k^G} . \tag{A.8}$$

This is essentially what we used in (2.20) to treat the case of fermion parity in the SYK model.

# B   Relation to other notions of approximate designs

In this appendix, we recall some basic inequalities that relate diamond definitions of approximate designs to the definition that we used based on the trace distance between moment superoperators. For more details, see [16, 40].

The diamond distance between completely positive maps can be defined as

$$\left\|\Phi_{\mathcal{E}}^{(k)} - \Phi_{\mathrm{Haar}}^{(k)}\right\|_\diamond = \sup_{\rho,n} \left\|\left(\left[\Phi_{\mathcal{E}}^{(k)} - \Phi_{\mathrm{Haar}}^{(k)}\right] \otimes 1_n\right)(\rho)\right\|_1 \tag{B.1}$$

where $\rho$ is a density matrix in $\mathcal{H}^{\otimes k} \otimes \mathcal{H}_n$. The diamond distance measures the distinguishability from the point of view of one-shot discrimination between quantum channels.

The diamond distance is bounded by the trace distance between superoperators

$$\frac{1}{d^k}\left\|\hat{\Phi}_{\mathcal{E}}^{(k)} - \hat{\Phi}_{\mathrm{Haar}}^{(k)}\right\|_1 \leqslant \left\|\Phi_{\mathcal{E}_t}^{(k)} - \Phi_{\mathrm{Haar}}^{(k)}\right\|_\diamond \leqslant \left\|\hat{\Phi}_{\mathcal{E}}^{(k)} - \hat{\Phi}_{\mathrm{Haar}}^{(k)}\right\|_1 . \tag{B.2}$$

The lower bound is obtained by choosing $n = d^k$ and $\rho = \dfrac{1}{d^k}|\infty\rangle\langle\infty|$. The upper bound is less trivial, it is obtained from the fact that the supremum in (B.1) is always achieved for $n \leqslant d^k$ and for a pure state $\rho = |\psi\rangle\langle\psi|$. We can use the operator representation $A_\Psi$ of this state to write

$$\left\|\Phi_{\mathcal{E}}^{(k)} - \Phi_{\mathrm{Haar}}^{(k)}\right\|_\diamond = \left\|\left(\left[\Phi_{\mathcal{E}}^{(k)} - \Phi_{\mathrm{Haar}}^{(k)}\right] \otimes 1_n\right)(|\psi\rangle\langle\psi|)\right\|_1 = \left\|A_\Psi \left(\hat{\Phi}_{\mathcal{E}}^{(k)} - \hat{\Phi}_{\mathrm{Haar}}^{(k)}\right) A_\Psi^\dagger\right\|_1 . \tag{B.3}$$

Using Hölder's inequality twice,

$$\left\|\Phi_{\mathcal{E}}^{(k)} - \Phi_{\mathrm{Haar}}^{(k)}\right\|_\diamond \leqslant \|A_\Psi\|_\infty \|A_\Psi^\dagger\|_\infty \left\|\left(\hat{\Phi}_{\mathcal{E}}^{(k)} - \hat{\Phi}_{\mathrm{Haar}}^{(k)}\right)\right\|_1 . \tag{B.4}$$

Using the monotonicity of the Schatten norms, and that $\|A_\Psi\|_2 = \|A_\Psi^\dagger\|_2 = 1$ from the purity of the

state, we get the upper bound in (B.2).

Therefore we see that our definition of approximate design is stronger than the diamond definition, and in the reverse direction, an $d^{-k}\varepsilon$-approximate $k$-design in diamond distance is an $\varepsilon$-design in trace distance between superoperators.

## C    A Brownian model with conservation of energy

In this appendix, we consider a Brownian model of the form (2.10) with the additional feature that it conserves the energy. We will explicitly show that for such a system, the ensemble of time-evolution unitaries $\mathcal{E}_t$ never becomes a $k$-design for any $k \geqslant 1$.

To construct the model, we select some Hamiltonian $H_0$. Now consider driving the system with this Hamiltonian, adding a time-dependent dimensionless coupling

$$H(t) = g(t)H_0\,. \tag{C.1}$$

In this model the time-evolution unitary corresponds effectively to the time-independent evolution with $H_0$,

$$U(t) = \mathsf{T}\,\exp\left(-i\int_0^t \mathrm{d}s\,H(s)\right) = e^{-it_g H_0}\,, \qquad t_g = \int_0^t \mathrm{d}s g(s)\,. \tag{C.2}$$

That is, the effect of the coupling is simply to explore different times on the submanifold generated by the fixed Hamiltonian $H_0$. The energy given by $H_0$ is obviously conserved, $[H_0, U(t)] = 0$.

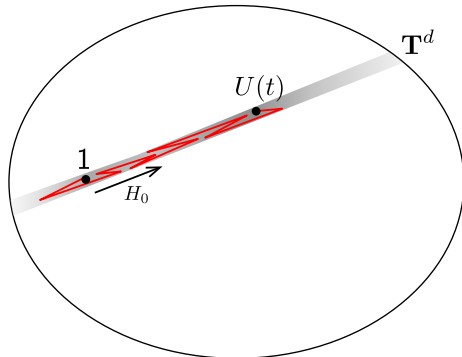

**Figure 6:** Cartoon of Brownian motion with the conservation of energy. The trajectory is restricted to a $\mathbf{T}^d \subset \mathsf{U}(d)$ submanifold. The thin red line represents the Brownian trajectory of $U(t) \in \mathcal{E}_t$ for a fixed realization of the coupling $g(t)$.

The coupling is taken to be random gaussian white-noise correlated

$$\mathbb{E}[g(t)] = 0\,, \qquad \mathbb{E}[g(t)g(0)] = t_0\delta(t)\,. \tag{C.3}$$

This induces a random gaussian variable for the time $t_g$ in (C.2) with $\mathbb{E}[t_g] = 0$ and $\mathbb{E}[t_g^2] = t_0 t$.

The moment superoperator $\hat{\Phi}_{\mathcal{E}_t}^{(k)}$ takes the form of an unnormalized thermal state (2.15), for the

effective Hamiltonian

$$H_k = t_0 \left( \sum_{r=1}^{k} H_0^r - H_0^{\bar{r}} \right)^2 . \tag{C.4}$$

For simplicity we assume that the Hamiltonian $H_0$ has time-reversal symmetry, so that $H_0^* = H_0$.

The effective Hamiltonian (C.4) can be explicitly diagonalized, in terms of the eigenbasis

$$\left| E_{I_k} \right\rangle \equiv \left| E_{i_1}, E_{i_{\bar{1}}}, ..., E_{i_1}, E_{i_{\bar{k}}} \right\rangle , \qquad H_k \left| E_{I_k} \right\rangle = E_{I_k} \left| E_{I_k} \right\rangle , \tag{C.5}$$

where

$$E_{I_k} = t_0 \left( E_{i_k} + ... + E_{i_k} - E_{i_{\bar{1}}} - ... - E_{i_{\bar{k}}} \right)^2 . \tag{C.6}$$

Let us assume that the spectrum of $H_0$ lacks of degeneracies and rational relations. This is what we expect in general if $H_0$ is chaotic [78]. At the level of the effective Hamiltonian (C.4) this means that the ground space $\mathrm{GS}_k$ is linearly generated by the eigenstates (C.5) where the individual energies are paired between forward an backward replicas

$$E_{i_{\bar{1}}} = E_{i_{\sigma(1)}}, ..., \; E_{i_{\bar{k}}} = E_{i_{\sigma(k)}} \qquad \sigma \in \mathrm{Sym}(k) . \tag{C.7}$$

There are in total $\dim(\mathrm{GS}_k) = d^k k!$ such linearly independent states for $k \leqslant d$. Note that, in particular, all of the states of the form (2.18) belong to this subspace, as expected from the general considerations of section 2.1.

According to (2.23) we get that the one-norm distance between the $k$-th moment superoperators is

$$Z_1(t) - Z_1(\infty) = k!(d^k - 1) + \sum_{E_{I_k} \neq 0} e^{-t E_{I_k}} . \tag{C.8}$$

For a general chaotic system, the level-spacing is $O(e^{-S(\bar{E})})$, where $\bar{E}$ denotes some average energy of the mircocanonical window and $S(\bar{E})$ represents the microcanonical entropy. The exponential decay in (C.8) is therfore very long lived, lasting for times $t \sim O(e^{2S(\bar{E})})$. More importantly, from the constant piece of (C.8), we see that the ensemble $\mathcal{E}_t$ never becomes an approximate $k$-design for any $k$.[18]

### Relation to spectral properties of $H_0$ and ergodicity

Note that our analysis above does not depend on the eigenbasis of $H_0$; it simply points to energy conservation as the source of non-generation of pseudorandomness. Indeed, from the conservation of $H_0$, all unitaries in $\mathcal{E}_t$ are constrained to a torus submanifold $\mathbf{T}^d \subset \mathsf{U}(d)$. This torus submanifold is

---

[18] This is also true for the weaker definition of approximate design in terms of the diamond distance, given the lower bound in (B.2).

defined by the unitaries which are diagonal in the energy basis of $H_0$,

$$U = \text{diag}(e^{i\theta_1}, ..., e^{i\theta_d}) \in \mathbf{T}^d, \qquad \theta_i \in (0, 2\pi]. \tag{C.9}$$

At long times the ensemble $\mathcal{E}_t$ simply becomes the ergodic cover of $\mathbf{T}^d$, in the sense that the measure defining the ensemble becomes the uniform measure $\frac{\mathrm{d}\theta_1...\mathrm{d}\theta_k}{(2\pi)^k}$ over $\mathbf{T}^d$.[19]

For example, using (2.32), we note that for this ensemble we can rewrite the partition function $Z_k(t)$ as a Brownian time-averaged version of the SFF

$$Z_k(t) = \int_{-\infty}^{\infty} \mathrm{d}s \, \frac{e^{-\frac{s^2}{2tt_0}}}{\sqrt{2\pi tt_0}} \, \text{SFF}(U(s))^k, \tag{C.10}$$

where the SFF was defined in (2.33).

At very long times the gaussian becomes uniform and we get

$$Z_k(\infty) = \overline{\text{SFF}^k}, \tag{C.11}$$

where the overbar represents the long-time average

$$\overline{f} \equiv \lim_{T \to \infty} \frac{1}{T} \int_{-T/2}^{T/2} \mathrm{d}s \, f(s). \tag{C.12}$$

For a Hamiltonian $H_0$ without rational relations on the spectrum, the long-time average is the same as the uniform average over $\mathbf{T}^d$,

$$Z_k(\infty) = \sum_{i_1,...,i_k,i_{\bar{1}},...,i_{\bar{k}}} \overline{e^{-it(E_{i_1}+...+E_{i_k}-E_{i_{\bar{1}}}-...-E_{i_{\bar{k}}})}} =$$

$$= \int \frac{\mathrm{d}^d\theta}{(2\pi)^d} \sum_{i_1,...,i_k,i_{\bar{1}},...,i_{\bar{k}}} e^{-i(\theta_{i_1}+...+\theta_{i_k}-\theta_{i_{\bar{1}}}-...-\theta_{i_{\bar{k}}})} = d^k k!. \tag{C.13}$$

We thus find that the origin of non-generation of randomness in this ensemble is the conservation of energy, which prevents $\mathcal{E}_t$ from exploring more than a measure zero submanifold $\mathbf{T}^d$ of the space of unitaries $\mathsf{U}(d)$. Therefore a typical unitary $U(t) \in \mathcal{E}_t$ will be far from random, even if $H_0$ is chosen in a very atypical way from the space of all Hamiltonians.

---

[19] Note that the uniform measure on $\mathbf{T}^d$ is the block-diagonal Haar measure studied in appendix A, with the peculiarity that the subspaces of fixed global charge (the energy in this case) are all one-dimensional. The hypothesis of ergodicity translates into the fact that there are no other invariant states to the ones found in appendix A; namely, that the charges of the different one-dimensional irreps have no rational relations and thus there are no additional singlets states in the tensor product representation.

# D    Other Brownian beta ensembles

In this section, we study the Brownian random matrix models defined in section 4 where the path integral measure $\mathcal{D}H(t)$ over Hamiltonians is chosen instead from the gaussian orthogonal ensemble (GOE) and gaussian symplectic ensemble (GSE).

## D.1    GOE

In systems with instantaneous time-reversal and rotation symmetry the Hamiltonian can be expanded as

$$H(t) = \sum_i x_{ii}(t)\Pi_{ii} + \sum_{i<j} x_{ij}(t)\frac{\Pi_{ij} + \Pi_{ji}}{\sqrt{2}}\,, \qquad \Pi_{ij} \equiv |i\rangle\langle j|\,. \tag{D.1}$$

The Brownnian GOE model is defined by (4.1) with the path integral measure

$$\mathcal{D}H(t) \propto \prod_i \mathcal{D}x_{ii}(t) \prod_{i<j} \mathcal{D}x_{ij}(t)\,. \tag{D.2}$$

With this choice, the real functions $x_{ij}(t)$ correspond to $K = \frac{d(d+1)}{2}$ real Brownian couplings with vanishing mean and variance

$$\mathbb{E}\left[x_{ij}(t)\right] = 0\,, \qquad \mathbb{E}\left[x_{ij}(t)x_{lm}(0)\right] = \frac{Jd}{K}\delta(t)\delta_{il}\delta_{jm}\,. \tag{D.3}$$

From the general considerations of section 2.1, the $k$-th moment superoperator $\hat{\Phi}_{\hat{\mathcal{E}}_t}^{(k)}$ of this ensemble can be written as the unnormalized thermal state (2.15) for the effective Hamiltonian

$$H_k = \frac{J}{2(d+1)}\sum_{i<j}\left(\sum_{r=1}^k \left[\Pi_{ij}^r + \Pi_{ji}^r - \Pi_{ij}^{\bar{r}} - \Pi_{ji}^{\bar{r}}\right]\right)^2 + \frac{J}{d+1}\sum_i \left(\sum_{r=1}^k \left[\Pi_{ii}^r - \Pi_{ii}^{\bar{r}}\right]\right)^2\,. \tag{D.4}$$

This operator can be recognized as

$$H_k = \frac{Jd}{d+1}\left(k + \sum_{r<s}\left(|\infty\rangle_{rs}\langle\infty|_{rs} + |\infty\rangle_{\bar{r}\bar{s}}\langle\infty|_{\bar{r}\bar{s}}\right) - \sum_{r,s}|\infty\rangle_{r\bar{s}}\langle\infty|_{r\bar{s}}\right) +$$

$$+ \frac{J}{d+1}\left(k + \sum_{r<s}(\text{SWAP}_{rs} + \text{SWAP}_{\bar{r}\bar{s}}) - \sum_{r,s}\text{SWAP}_{r\bar{s}}\right)\,. \tag{D.5}$$

In this form, it is manifest that the effective Hamiltonian $H_k$ is invariant under $\mathsf{SO}(d)$ subgroup of the unitary transformations that rotate the reference basis, while leaving invariant time-reversal symmetry. Note that since the symmetry is in this case a subgroup of $\mathsf{SU}(d)$, the effective Hamiltonian (D.5) is less restricted than (4.7). In particular, $\text{SWAP}_{r\bar{s}}$, $|\infty\rangle_{rs}\langle\infty|_{rs}$ and $|\infty\rangle_{\bar{r}\bar{s}}\langle\infty|_{\bar{r}\bar{s}}$ are only invariant under the $\mathsf{SO}(d)$ subgroup of transformations of the reference basis, and not under the full $\mathsf{SU}(d)$ group.

*Time to 1-design*

The effective Hamiltonian (D.4) for $k = 1$ is

$$H_1 = \frac{Jd}{d+1} \left(1 - |\infty\rangle_{1\bar{1}} \langle\infty|_{1\bar{1}}\right) + \frac{J}{d+1} \left(1 - \mathrm{SWAP}_{1\bar{1}}\right), \tag{D.6}$$

Let us consider a general state

$$|\psi\rangle_{1\bar{1}} = \sum_{ij} \psi_{ij} |i, j\rangle_{1\bar{1}}. \tag{D.7}$$

The eigenspaces of $H_1$ fall into irreps of $\mathsf{SO}(d)$ in the tensor product

$$\mathbf{d} \otimes \mathbf{d} = \mathbf{1} \oplus \mathbf{A}_2 \oplus \mathbf{S}_2 \tag{D.8}$$

where $\mathbf{1}$ is the singlet ($\psi_{ij} \propto \delta_{ij}$), $\mathbf{S}_2$ is the rank 2 symmetric traceless ($\psi_{ij} = \psi_{(ij)}$, $\sum_i \psi_{ii} = 0$) and $\mathbf{A}_2$ is the rank 2 antysymmetric ($\psi_{ij} = \psi_{[ij]}$). These eigenspaces have corresponding eigenvalue $\{0, J\frac{d}{d+1}, J\frac{d+2}{d+1}\}$, respectively. The spectral gap is therefore $\Delta_1 = J\frac{d}{d+1}$.

The trace distance to 1-design in this ensemble is then

$$Z_1(t) - Z_1(\infty) = e^{-\Delta_1 t} \left[ \left(\frac{d(d+1)}{2} - 1\right) + \frac{d(d-1)}{2} e^{\frac{Jt}{d+1}} \right]. \tag{D.9}$$

The time to 1-design is then

$$t_1 \approx \frac{1}{J} \left(2 \log d + \log \varepsilon^{-1}\right). \tag{D.10}$$

*Time to 2-design*

For illustrative purposes, we shall explicitly study the $k = 2$ effective Hamiltonian

$$H_2 = \frac{Jd}{d+1} \left(2 + |\infty\rangle_{12}\langle\infty|_{12} + |\infty\rangle_{\bar{1}\bar{2}}\langle\infty|_{\bar{1}\bar{2}} - |\infty\rangle_{1\bar{1}}\langle\infty|_{1\bar{1}} - |\infty\rangle_{2\bar{2}}\langle\infty|_{2\bar{2}} - |\infty\rangle_{1\bar{2}}\langle\infty|_{1\bar{2}} - |\infty\rangle_{2\bar{1}}\langle\infty|_{2\bar{1}}\right) +$$

$$+ \frac{J}{d+1} \left(2 + \mathrm{SWAP}_{12} + \mathrm{SWAP}_{\bar{1}\bar{2}} - \mathrm{SWAP}_{1\bar{1}} - \mathrm{SWAP}_{2\bar{2}} - \mathrm{SWAP}_{1\bar{2}} - \mathrm{SWAP}_{2\bar{1}}\right). \tag{D.11}$$

Consider a general state

$$|\psi\rangle_{1\bar{1}2\bar{2}} = \sum_{ijkl} \psi_{ijkl} |i, j, k, l\rangle_{1\bar{1}2\bar{2}}. \tag{D.12}$$

To construct the irreducible representations of $\mathsf{SO}(d)$ in this Hilbert space, we can start from the irreps

of $\mathsf{GL}(d)$, which can be constructed from the standard Young diagrammatics

$$\square^{\otimes 4} = \square\square\square\square \oplus \square \oplus \square\square\square^{\times 3} \oplus \square^{\times 3} \oplus \square\square^{\times 2}. \tag{D.13}$$

These diagrams define a set of Young projectors onto invariant subspaces of $\mathsf{GL}(d)$, which can be easily read by labelling the boxes with the $ijkl$ indices and following the standard rules (see e.g. [56]).

The first two tableaux in (D.13) correspond to the totally symmetric and antisymmetric tensors, $\psi_{ijkl} = \psi_{(ijkl)}$ and $\psi_{ijkl} = \psi_{[ijkl]}$, respectively. The next two correspond to mixed representations, where the Young projector is more complicated. The last representation corresponds to a tensor with the symmetries of the Riemann tensor, $\psi_{ijkl} = \psi_{[ij]kl} = \psi_{ij[kl]}$ and $\psi_{i[jkl]} = 0$. Given this decomposition, to construct irreps of $\mathsf{SO}(d)$ we must separate these tensors into their traceless parts and consider their traces separately. By the Schur-Weyl duality the decomposition into irreps of $\mathsf{SO}(d)$ will generate irreps of $\mathrm{Sym}(2k)$ as well.

The simplest example comes from the rank 4 totally antisymmetric representation $\mathbf{A}_4$, which is traceless and thus irreducible for $d > 8$ (for $d = 8$ it reduces into self-dual and anti self-dual part, while for $d < 8$ it is dual to an antisymmetric representation of rank $d-4 < 4$). This subspace ($\psi_{ijkl} = \psi_{[ijkl]}$) has dimension $\binom{d}{4}$ and energy $2J\frac{d+2}{d+1}$. The rest of the representations in (D.13) are reducible.

It is clear from (D.13) that there will be three singlets. One of them will come from the totally symmetric representation of $\mathsf{GL}(d)$; the other two will come from the two 'Riemann tensors' (these encode the analog of the Ricci scalar). In the space of singlets, there will be two ground states $|W_\sigma\rangle$ for $\sigma = e, (12)$ and an excited state, $\psi_{ijkl} \propto \delta_{ik}\delta_{ij}$ with energy $4J$. Therefore, the first difference with the GUE case of section 4 is that the irrep of $\mathsf{SO}(d)$ does not determine the energy totally. The reason is that the Hamiltonian (D.11) is not invariant under the full permutation group $\mathrm{Sym}(2k)$; it is only invariant under a subgroup $\mathrm{Sym}(k) \times \mathrm{Sym}(k)$ that swaps the forward and backward replicas separately.

Moreover, these three representations will yield a traceless symmetric rank 2 tensor $\mathbf{S}_2$ each. This will leave the rank 4 totally symmetric traceless $\mathbf{S}_4$ (of dimension $\binom{d+3}{4} - \binom{d+1}{2}$) and two rank 4 representations $\mathbf{W}_4$ with the symmetries of the Weyl tensor (of dimension $\frac{d(d+1)(d+2)(d-3)}{12}$). On the other hand, the mixed representations of (D.13) yield a rank 2 antisymmetric tensor $\mathbf{A}_2$ each, while the third representation in (D.13) will yield a rank 2 symmetric traceless tensor $\mathbf{S}_2$ as well. The remaining representation are traceless rank 4 mixed representations, $\mathbf{M}_{(3,1)}$ and $\mathbf{M}_{(1,3)}$.

Therefore, the eigenspaces of the effective Hamiltonian will respect the decomposition

$$\mathbf{d} \otimes \mathbf{d} \otimes \mathbf{d} \otimes \mathbf{d} = \mathbf{1}^{\times 3} \oplus \mathbf{A}_2^{\times 6} \oplus \mathbf{S}_2^{\times 6} \oplus \mathbf{M}_{(3,1)}^{\times 3} \oplus \mathbf{M}_{(1,3)}^{\times 3} \oplus \mathbf{W}_4^{\times 2} \oplus \mathbf{A}_4 \oplus \mathbf{S}_4. \tag{D.14}$$

Note that by inspection of (D.11) the traceless representations all have energy $E \approx 2J$ if we neglect energy separations of $O(J/d)$. Four of the $\mathbf{A}_2$ and $\mathbf{S}_2$ (the ones corresponding to single-replica excitations

on top of the ground states, e.g. $\psi_{ijkl} = \delta_{ij}\phi_{kl}$, with $\phi_{kl}$ traceless) will have energies $E \approx \Delta_2 \approx J$.

Since most of the states lie at $E \approx 2J$, we can approximate the trace distance to 2-design by

$$Z_2(t) - Z_2(\infty) \approx 4(d^2 - 1)e^{-Jt} + (d^4 - 4d^2 - 2)e^{-2Jt} , \tag{D.15}$$

which leads to a time to 2-design of

$$t_2 \approx \frac{1}{J} \left( \log(d^2 - 1) + 2\log 2 + \log \varepsilon^{-1} \right) , \tag{D.16}$$

exactly like in the GUE case.

*Time to k-design*

For general $k \leqslant d$, the eigenspaces of $H_k$ in (D.4) will admit the decomposition

$$\underbrace{\mathbf{d} \otimes ... \otimes \mathbf{d}}_{2k} = \mathbf{1}^{\times \frac{(2k)!}{2^k}} \oplus ... \tag{D.17}$$

Now there will be more singlets than in the GUE case, corresponding to the total number of pairings of the $2k$ replicas. However, only $k!$ of these will be ground states, for the general reasons stated above. Moreover, the first excited states will correspond to single replica excitations (either $\mathbf{S}_2$ or $\mathbf{A}_2$) on top of each ground state. There will be a total of $N_* = k!k(d^2 - 1)$ first excited states. Note that the traceless representations (most of the Hilbert space) has energy $E \approx kJ$ for $k/d \ll 1$. Therefore, this leads to a time to $k$-design

$$t_k \sim \frac{1}{J} \left( k \log k - k + \log k + \log(d^2 - 1) + \log \varepsilon^{-1} \right) , \tag{D.18}$$

which up to logarithmic factors and subleading terms is linear in $k$, in complete agreement with the GUE case.

## D.2   GSE

For even $d$, in a system with time reversal symmetry but without rotational symmetry, the time-dependent Hamiltonian can be expressed as

$$H(t) = \sum_i a_{ii}(t)\Pi_{ii} + \sum_{i<j} \left( a_{ij}(t)\frac{\Pi_{ij} + \Pi_{ji}}{\sqrt{2}} + (b_{ij}(t)\mathbf{i} + c_{ij}(t)\mathbf{j} + d_{ij}(t)\mathbf{k})\frac{\Pi_{ij} - \Pi_{ji}}{\sqrt{2}} \right) , \tag{D.19}$$

where $i, j = 1, ..., \frac{d}{2}$ and the matrix entries are quaternions $H_{ij} = a_{ij} + b_{ij}\mathbf{i} + c_{ij}\mathbf{j} + d_{ij}\mathbf{k}$, represented as $2 \times 2$ matrices and $\Pi_{ij} \equiv |i\rangle\langle j|$. The Brownian GSE model is defined by (4.1) with the path integral measure

$$\mathcal{D}H(t) \propto \prod_i \mathcal{D}a_{ii}(t) \prod_{i<j} \mathcal{D}a_{ij}(t)\mathcal{D}b_{ij}(t)\mathcal{D}c_{ij}(t)\mathcal{D}d_{ij}(t) . \tag{D.20}$$

In the Brownian GSE model, the functions $\{a_{ij}(t), b_{ij}(t), c_{ij}(t), d_{ij}(t)\}$ are taken as $K = \frac{d(d-1)}{2}$ white-noise correlated gaussian random couplings with vanishing mean and variance

$$\mathbb{E}\left[a_{ij}(t)a_{lm}(0)\right] = \mathbb{E}\left[b_{ij}(t)b_{lm}(0)\right] = \mathbb{E}\left[c_{ij}(t)c_{lm}(0)\right] = \mathbb{E}\left[d_{ij}(t)d_{lm}(0)\right] = \frac{Jd}{K}\delta(t)\delta_{il}\delta_{jm}\,. \qquad \text{(D.21)}$$

From the general considerations of section 2.1, the $k$-th moment superoperator $\hat{\Phi}_{\mathcal{E}_t}^{(k)}$ of this ensemble can be written as the unnormalized thermal state (2.15) for the effective Hamiltonian

$$H_k = \frac{J}{2(d-1)}\sum_{i<j}\left[\left(\sum_{r=1}^{k}\Pi_{ij}^r + \Pi_{ji}^r - \Pi_{ij}^{\bar{r}} - \Pi_{ji}^{\bar{r}}\right)^2 - 3\left(\sum_{r=1}^{k}\Pi_{ij}^r - \Pi_{ji}^r + \Pi_{ij}^{\bar{r}} - \Pi_{ji}^{\bar{r}}\right)^2\right] +$$

$$+ \frac{J}{d-1}\sum_{i}\left(\sum_{r=1}^{k}\left[\Pi_{ii}^r - \Pi_{ii}^{\bar{r}}\right]\right)^2\,. \qquad \text{(D.22)}$$

We can reorganize this operator as

$$H_k = \frac{J}{d-1}\left(2k(d-1) - d\sum_{r<s}(|\infty\rangle_{rs}\langle\infty|_{rs} + |\infty\rangle_{\bar{r}\bar{s}}\langle\infty|_{\bar{r}\bar{s}}) - 2d\sum_{r,s}|\infty\rangle_{r\bar{s}}\langle\infty|_{r\bar{s}} + \right.$$

$$\left. + 4\sum_{r<s}(\text{SWAP}_{rs} + \text{SWAP}_{\bar{r}\bar{s}}) + 2\sum_{r,s}\text{SWAP}_{r\bar{s}}\right)\,. \qquad \text{(D.23)}$$

which is invariant under the symplectic subgroup $\mathsf{Sp}(d)$ of unitary transformations of the reference basis. In this form, the effective Hamiltonian (D.23) is only manifestly invariant under a $\mathsf{SO}(\frac{d}{2})$ subgroup, and it corresponds to the identity in the $2 \times 2$ factor. Therefore, the diagonalization of such Hamiltonian proceeds in analogy to the GOE case, in the replicated Hilbert space where each factor has dimension $\frac{d}{2}$. For this reason, we shall not attempt to repeat the previous analysis here. The only subtlety in the GSE case is that there will be an additional fourfold degeneracy on each factor, given that every Hamiltonian of the ensemble (D.19) respects the factorization of the Hilbert space. Thus, the ensemble $\mathcal{E}_t$ will only reach the factorized Haar ensemble at infinite times.

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
