# Peer review of "Complexity is not Enough for Randomness"

_SciPost Physics_

## Round 1 · Referee Report · Anonymous (Referee 1) · 2024-7-24

Report
in a Hamiltonian system? The authors propose that it is indeed hard: the time it would take scales
exponentially with a system size. To corroborate this statement they present several universal arguments in the Introduction and also study several physical systems: the Brownian Sachdev--Ye--Kitaev (SYK) model, Brownian random
matrix and a diffusion on the unitary group.
The paper is very-well written, the physical question it addresses is important and I generally agree with the conclusion, so I highly recommend the paper for the publication. However, I would like the authors to address a few questions.
The main one is related to the design time. Eqns. (3.37), (4.29) for the design time contain 1/J factor.
Suppose that my sole goal is to generate a Haar random matrix using the SYK model. Would it be a legit thing to do
to make J very large (exponential in the system size) in order to achieve Haar randomness at times of order 1?
Also do I understand correctly that the authors would call such scaling unphysical because the two-point function
would decay on a time-scale of order inverse exponential system size?
Also I have a few less important questions:
I agree with most of the arguments from the Introduction about atypicality of $H_{Haar}$.
However, I am confused by the following statement on p.4: "On the other hand, the Hamiltonian H_{Haar} has approximately uniform spectrum on an interval". I thought that the eigenphases of a Haar random matrix do
exhibit repulsion - e.g. eq. (2.35) from the paper.
p.7: "...the growth in circuit complexity is expected to
generally be hyperfast in non-local systems. "
Does "hyperfast" mean that the complexity is saturated at times of order 1?
In this paper \epsilon-approximate k-design is defined through Schatten 1-norm of the corresponding
superoperator - eq. (2.5) . What is the motivation for using this norm? Schatten 1-norm is very useful for bounding
the correlation functions via the Holder inequality,
but as the authors point out in Appendix B, the diamond norm is generally smaller.
I am wondering if the design times will be significantly affected by the choice of the norm.
Finally, I noticed one typo:
Bottom of p.5: "non-trivial part the becomes to".
Recommendation
Ask for minor revision

---

## Round 1 · Referee Report · Anonymous (Referee 2) · 2024-8-26

Report
The introduction provides the necessary motivation and background. The technical sections are clear and present calculations to an appropriate level of detail. The appendices provide relevant additional details.
Given the novelty and significance of this work, I recommend this paper for publication with a few minor changes listed below.
Requested changes
-
On page 12, the authors have some general comments about when $\text{V}_k = \text{GS}_k$ pertaining to the lack of clusters and global symmetries. This is a fairly important point and it would be helpful to explain the intuition behind this claim in greater detail.
-
On page 17, below (3.7) the authors should define $Q^r$.
-
On page 35, (4.44) $\varepsilon^{-1}$ should be changed to $\log \varepsilon^{-1}$.
-
On page 44, below (C.6) the authors assume that $H_0$ lacks degeneracies and rational relations. If this assumption is violated, I would expect $\text{dim GS}_k$ to be larger. This would still imply that $\mathcal{E}_t$ never becomes a $k$-design. Some clarification to this end would be helpful.
Recommendation
Ask for minor revision

---

## Editorial Decision

unknown